# A modular toolbox to generate complex polymeric ubiquitin architectures using orthogonal sortase enzymes

Maximilian Fottner[1,2], Maria Weyh [1], Stefan Gaussmann [3,4], Dominic Schwarz[1], Michael Sattler[3,4] & Kathrin Lang [2✉]

The post-translational modification of proteins with ubiquitin (Ub) and Ub-like modifiers (Ubls) represents one of the most important regulators in eukaryotic biology. Polymeric Ub/Ubl chains of distinct topologies control the activity, stability, interaction and localization of almost all cellular proteins and elicit a variety of biological outputs. Our ability to characterize the roles of distinct Ub/Ubl topologies and to identify enzymes and receptors that create, recognize and remove these modifications is however hampered by the difficulty to prepare them. Here we introduce a modular toolbox (Ubl-tools) that allows the stepwise assembly of Ub/Ubl chains in a flexible and user-defined manner facilitated by orthogonal sortase enzymes. We demonstrate the universality and applicability of Ubl-tools by generating distinctly linked Ub/Ubl hybrid chains, and investigate their role in DNA damage repair. Importantly, Ubl-tools guarantees straightforward access to target proteins, site-specifically modified with distinct homo- and heterotypic (including branched) Ub chains, providing a powerful approach for studying the functional impact of these complex modifications on cellular processes.

[1] Department of Chemistry, Lab for Synthetic Biochemistry, Technical University of Munich, Institute for Advanced Study, TUM-IAS, Lichtenberg Str. 4, 85748 Garching, Germany. [2] Laboratory of Organic Chemistry, Department of Chemistry and Applied Biosciences, ETH Zürich, Vladimir-Prelog-Weg 3, 8093 Zurich, Switzerland. [3] Bavarian NMR Center, Department of Chemistry, Technical University of Munich, Lichtenberg Str. 4, 85748 Garching, Germany. [4] Institute of Structural Biology, Helmholtz Zentrum München, Ingolstädter Landstrasse 1, 85764 Neuherberg, Germany. ✉email: kathrin.lang@org.chem.ethz.ch

Ubiquitylation regulates numerous cellular processes, including protein degradation, DNA repair, receptor transport, and viral infection[1,2]. During ubiquitylation the small protein ubiquitin (Ub) is attached via its C-terminus to a lysine residue within a substrate protein by forming an isopeptide bond. The functional diversity of ubiquitylation is achieved by the ability of Ub to form different types of conjugates[3]. Mono-ubiquitylation typically alters a target protein's interaction landscape and has been shown to be involved in regulation of DNA repair and receptor endocytosis[4]. In addition, cells can assemble polymeric Ub chains, in which Ub monomers are connected through an isopeptide bond between eight possible attachment sites on one Ub monomer (the N-terminus M1 and lysine residues K6, K11, K27, K29, K33, K48, and K63) and the C-terminus of the next Ub monomer[1]. Homotypic chains are characterized by a single predominant linkage, such as in K48-linked conjugates that promote proteasomal degradation[5] and in K63-chains that are involved in NF-κB[6] signaling and DNA damage response[7]. Recent biochemical studies have shown that in addition to homogenously linked Ub chains, polymers containing heterotypic linkages that adopt mixed or branched topologies also control various cellular processes[3,8]. Mixed chains are composed of different linkages, but each Ub is modified with only one other Ub molecule. If instead a single Ub subunit within a chain is modified with two or more Ub molecules at a time, branched chain structures are generated. Distinct cellular functions have been assigned to K11/K48-, K29/K48-, K48/K63-, and M1/K63-branched conjugates and it is likely that differences in the number and/or sequence of branches might affect the outcome of these modifications[8]. It has for example been shown that many branched chains elicit particularly efficient proteasomal degradation[9,10]. A further level of complexity in Ub signaling is added by occurrence of so-called hybrid chains, heterologous polymers that consist of mixed Ub and Ubiquitin-like-modifier (Ubl) subunits[11]. Hybrid Ub-SUMO (small ubiquitin modifier) chains are supposed to play a major role in DNA repair[12]. Yet, due to the difficulty in creating defined complex Ub/Ubl architectures, the physiological roles of most chain types remain poorly understood. Approaches to access defined and complex Ub/Ubl chains include chemical protein synthesis[13] or strategies based on cysteine modifications and chemoselective chemistries (e.g. Cu(I)-azide-alkyne cycloaddition[14] and thiol-ene chemistry[15]) to connect Ub monomers via either native or non-native isopeptide bonds. Many of these methods do however require advanced chemical expertise and manipulations and are therefore not easily implementable in typical biology research labs. Furthermore, they are in most cases restricted to simple refoldable target proteins that do not contain cysteines. Alternatively, sophisticated enzymatic approaches using a combination of linkage-specific conjugating enzymes and Ub mutants have guaranteed access to simple Ub chains[16,17], but it is difficult to extend these protocols to more complex Ubl topologies and it is so far not possible to site-specifically ligate Ubl chains to a certain substrate protein.

To overcome some of the limitations faced in generating Ub/Ubl-conjugates, we have recently developed a chemoenzymatic approach that allows site-specific ubiquitylation and SUMOylation of substrate proteins (sortylation)[18]. Sortylation is based on genetic incorporation of the unnatural amino acid GGK (Fig. 1a) into a protein of interest (POI) that subsequently serves as a substrate for sortase-mediated transpeptidation with a Ubl variant bearing a sortase recognition motif within its C-terminus, resulting in a Ubl-POI conjugate linked via a native isopeptide bond. In contrast to reported methods, sortylation allows the generation of defined Ubl-POI conjugates under native conditions and consequently the modification of nonrefoldable, multimeric proteins both in vitro and in cellulo. Nevertheless, sortylation is limited to monoubiquitylation/monoSUMOylation events.

Here, we present a modular toolbox for accessing complex polymeric Ub/Ubl topologies, including heterotypic, hybrid, and branched chains. Our approach is based on identification of a set of sortase enzymes with affinity and specificity for unique and orthogonal recognition motifs. These sortase variants can be used iteratively to ligate bifunctional Ub and Ubl monomers bearing the respective recognition motifs within their C-termini and a GGK-residue at a user-defined position in a programmable and flexible fashion. We name this approach "**Ubl-to**pologies via **o**rthogonal **s**ortylation" (Ubl-tools, Fig. 1b). We demonstrate the generality of Ubl-tools by building differently linked diUb-SUMO2 hybrid chains. This allows us to investigate their involvement in DNA double strand break (DSB) repair by examining their interaction with the BRCA1-A adaptor protein Rap80[12], and we suggest a previously uncharacterized binding mode of a distinct diUb-SUMO hybrid chain to Rap80. Furthermore, we show that Ubl-tools can be combined with enzymatic Ub assembly to site-specifically charge complex Ub chains, such as K48-linked tetraUb or linkage-defined branched Ub oligomers—signals known to play fundamental roles in proteasomal degradation—onto target proteins[9,19]. The modular concept of Ubl-tools combined with its ease of implementation makes it a valuable approach for generating otherwise inaccessible Ub/Ubl architectures and we envision it will open up new opportunities for studying the functional impact of these complex types of modifications on critical cellular processes.

## Results

**Identification of orthogonal sortase enzymes for iterative sortylation.** Towards our goal to build complex Ubl topologies, we set out to identify sortase variants with specificity for distinct and orthogonal recognition motifs that are suitable for iterative sortylation. Sortase A (SrtA) enzymes from *Staphylococcus aureus* and *Streptococcus pyogenes* have been used previously in an iterative manner for double modification of proteins[20] and for protein circularization[21]. Their orthogonality, however, stems from the different nature of the respective acceptor nucleophiles: while SrtA$_{aur}$ uses a diglycine as minimal acceptor nucleophile, SrtA$_{pyo}$ is dependent on a dialanine moiety[22,23], making it inept for sortylation. We therefore envisioned to investigate the orthogonality of evolved SrtA$_{aur}$ variants with reprogrammed recognition motifs for Ubl-tools. Srt2A recognizes the LAXTG motif (X = any amino acid) and Srt4S shows affinity for LPXSG[24]. Both variants are derived from Srt5M, a catalytically improved variant of the wild type (wt) SrtA$_{aur}$ that recognizes the archetypical LPXTG motif[25]. To test the suitability of these engineered sortase variants for Ubl-tools, we designed an orthogonality assay aimed at interrogating their proficiency for hydrolyzing the individual recognition motifs in a diUb context. For accessing differently linked diUbs, we expressed individual Ub variants displaying the recognition motifs for Srt2A, Srt4S, and Srt5M in their C-termini and incubated them in the presence of a Ub variant bearing GGK at position K6 (Ub-K6GGK) and the respective sortase enzymes (Supplementary Fig. 1). For enhanced sensitivity we introduced a leucine spacer preceding the sortase recognition motif, creating Ub(LAT), Ub(LPT), and Ub(LPS) (Fig. 1c), as this increases accessibility of sortase to the Ub C-terminus and therefore both sortylation as well as hydrolysis of the corresponding motif upon prolonged sortase incubation[18]. The three purified K6-diUb variants were incubated individually with the three different sortases (Fig. 1d and Supplementary Fig. 2). As expected, prolonged incubation of a diUb variant with its own sortase enzyme, i.e. the sortase variant that was used to assemble it and therefore displays

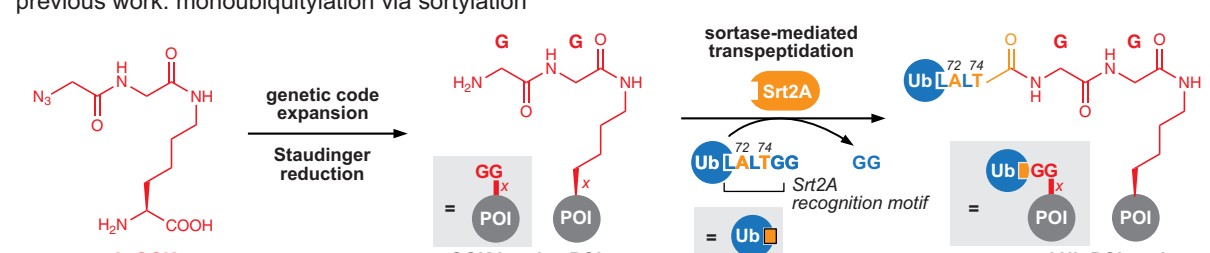

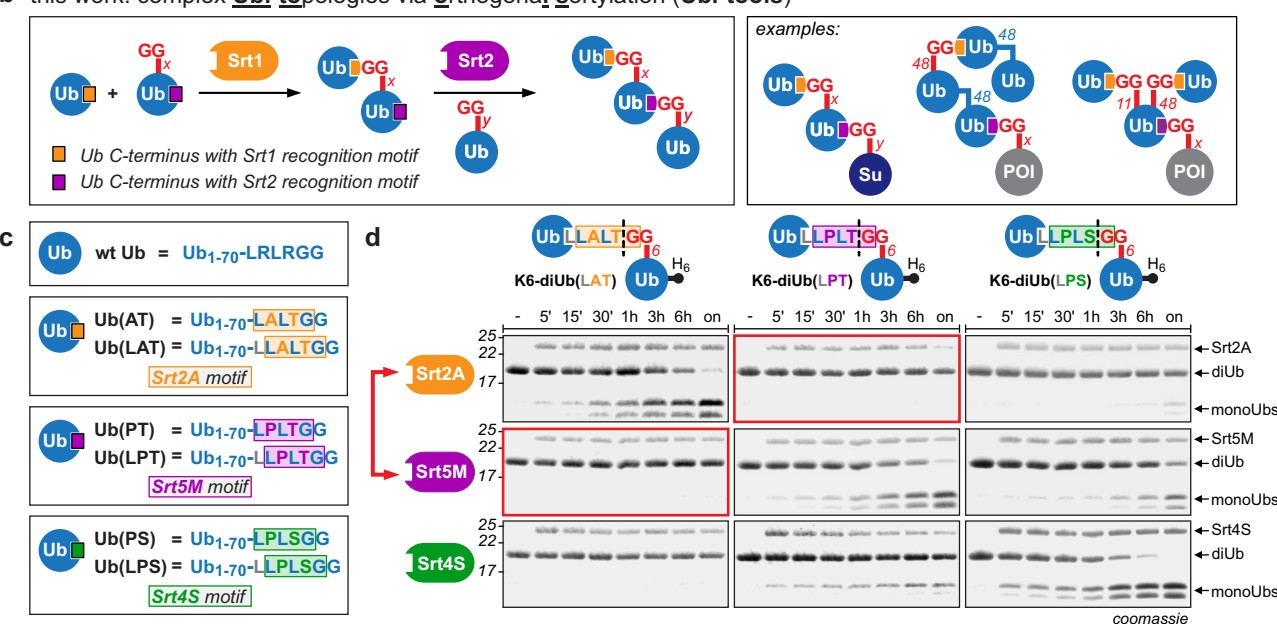

**Fig. 1 Access to defined Ub/Ubl topologies via orthogonal sortylation (Ubl-tools). a** Sortylation allows site-specific monoubiquitylation of a protein of interest (POI) and relies on site-specific incorporation of AzGGK, Staudinger reduction to access GGK-modified proteins and sortase-mediated transpeptidation. The resulting Ub-POI conjugates display a native isopeptide bond and two point mutations in the Ub C-terminus (R72A and R74T). **b** Ubl-tools expands sortylation to complex Ub/Ubl topologies by implementing a pair of orthogonal sortases that enables iterative sortase-mediated transpeptidation (left). Examples of Ub/Ubl architectures accessible via Ubl-tools (right). **c** Overview and nomenclature of different Ub variants bearing distinct sortase recognition motifs in their C-termini. **d** Orthogonality assay using differently linked diUbs. Each of the three sortase-generated diUbs was incubated individually with all three different sortase enzymes. Samples taken at the denoted time points were analysed by SDS-PAGE. All sortase variants exhibit on-target hydrolysis activity on a diUb displaying their own recognition motif at the linkage site. Srt2A as well as Srt5M show off-target reactivity for the Srt4S recognition motif. Furthermore, Srt4S hydrolyzes the Srt2A recognition motif. Srt2A and Srt5M are bidirectionally orthogonal towards their corresponding recognition motifs and can be used as orthogonal sortases (red rectangles and red arrow). Consistent results were obtained over at least three replicate experiments. Source data are provided as a Source Data file.

the target recognition motif, led to on-target diUb hydrolysis yielding the two corresponding Ub monomers. Furthermore, Srt5M hydrolyzed the diUb variant bearing the Srt4S target motif and vice versa, and Srt2A showed off-target hydrolysis for the Srt4S motif-linked diUb. In contrast, Srt4S did not cleave the diUb harboring the Srt2A-motif linker. Excitingly, diUb bearing the Srt5M motif in its linker region was completely refractory to hydrolysis by Srt2A and vice versa, meaning that Srt5M and Srt2A constitute a bidirectionally orthogonal sortase pair that can be used iteratively to ligate Ub and Ubl monomers (Fig. 1d and Supplementary Fig. 2).

Having established an orthogonal sortase pair for Ubl-tools, we first examined if Ub variants with correspondingly modified C-termini were still substrates for deubiquitylases (DUBs). While we observed complete cleavage of the C-terminal hexa-histidine $H_6$-tag upon incubation of wt Ub-$H_6$ with the catalytic domain of USP2 or with UCHL3, Ub variants bearing a C-terminal $H_6$-tag succeeding the Srt2A ('AT' or 'LAT') or Srt5M ('PT' or 'LPT')

motif were refractory to DUB cleavage (Supplementary Fig. 3). Resistance to DUB hydrolysis constitutes an important feature for Ubl-tools as it allows to use the generated Ub/Ubl topologies for identifying Ub chain-specific interactor proteins in cell lysates and provides valuable tools for interrogating cell-signaling pathways. A crucial determinant for Ub-mediated cellular signaling consists in the ability of effector proteins to convert distinct Ub patterns into specific functional outcomes[1,26]. To experimentally validate the functional and structural integrity of our sortase-generated linkages, we set out to build K63- and K48-linked diUbs bearing the different Srt2A and Srt5M motifs in the linker region connecting the two Ub monomers (diUb(AT), diUb(LAT), diUb(PT), and diUb(LPT)) and tested whether they are selectively recognized by specific Ub-binding domains (UBDs). First, we incubated these sortase-generated variants as well as their respective wt counterparts (i.e. wt K63-diUb and wt K48-diUb) with a K63-linkage-specific antibody (Fig. 2a and Supplementary Fig. 4a). All K63-linked diUbs were recognized to

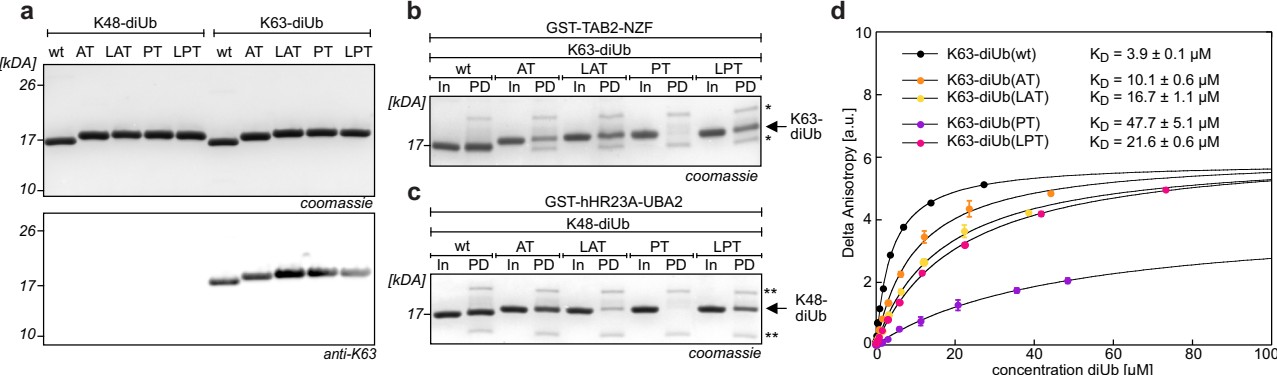

**Fig. 2 Functional characterization of sortase-generated diUb building blocks used for Ubl-tools. a** SDS-PAGE of K48- and K63-linked diUbs displaying different sortase motifs at their linkage sites (top: Coomassie-stained SDS-PAGE gels, bottom: western blot (WB) using an anti-K63-linkage-specific antibody). **b** SDS-PAGE analysis of pull-down (PD) experiments using differently linked K63-diUbs and GST-fused TAB2-NZF. Single asterisk (*) marks impurities present in GST-TAB2-NZF. **c** SDS-PAGE analysis of PD experiments using differently linked K48-diUbs and GST-fused hHR23A-UBA2 (bottom). Double asterisks (**) marks impurities present in GST- hHR23A-UBA2. In: input of differently linked K48- or K63-diUbs, PD: pull-down. Full gels can be found in Supplementary Fig. 4. **d** Determination of binding constants of differently linked K63-diUbs towards fluorescently labeled Rap80-tUIMs$_{(79-124)}$ using fluorescence anisotropy. Delta anisotropy was plotted against diUb concentration and fitted with a single-site binding model to determine $K_D$ values. Average values and error bars (s.d.) were calculated from three different experiments ($n = 3$). All data processing was performed using Kaleidagraph software (Synergy Software, Reading, UK) or Prism 9.2.0 (GraphPad Software, LLC). Consistent results were obtained over at least three replicate experiments. Source data are provided as a Source Data file.

a similar extent to natively linked K63-diUb in anti-K63 western blots, indicating that AT, LAT, PT, or LPT substitutions in the linker region do not interfere with binding to the linkage-specific antibody. As expected, all K48-linked diUbs did not bind to the K63-linkage-specific antibody. Next, we probed K63-linked diUbs in in vitro pull-down (PD) assays with protein kinase TAK1 adaptor subunit TAB2, which contains an Npl4 zinc finger (NZF) UBD that specifically senses K63-linked chains (Fig. 2b and Supplementary Fig. 4b)[26–28]. K63-diUb(AT), K63-diUb(LAT), and K63-diUb(LPT) retained the ability to bind TAB2-NZF, while K63-linked diUb(PT) containing the PT linker between the two Ub monomers showed compromised binding, indicating that the R72P substitution might be less optimal for mimicking wt K63-diUb behavior. Similarly, also the more compact sortase-generated K48-diUbs displaying AT, LAT, and LPT substitutions retained their ability to bind the designated Ub-associated (UBA) domain of proteasomal shuttling factor hHR23A-UBA2 in in vitro PD assays (Fig. 2c and Supplementary Fig. 4c), while K48-diUb(PT) failed to properly bind to hHR23A-UBA2[29]. To study binding properties of different sortase-generated diUbs in more quantitative terms, we determined binding constants of differently linked K63-diUbs and a fluorescently labeled Rap80 construct that harbors K63-sensitive tandem Ub-interacting motifs (tUIMs) via fluorescence anisotropy (Rap80-tUIMs$_{(79-124)}$, Fig. 2d)[30,31]. For all four sortase-generated K63-diUbs we measured distinct binding affinities. $K_D$s for AT-, LAT-, and LPT-linked diUbs were slightly lower as for wt K63-diUb (approximately two-fold reduction for diUb(AT) and four- to five-fold reduction for diUb(LAT) and diUb(LPT), respectively). K63-diUb(PT) displayed a 10-fold lower binding affinity towards Rap80-tUIMs (Fig. 2d), confirming our previous observation that the proline mutation at position 72 might give diUbs an unusual conformational rigidity. This indicates that the PT linker may be less optimal than the other investigated sortase linkers for recognition by some UBDs and that it might therefore be beneficial to introduce the leucine spacer amino acid to resemble more wt-like behavior. Nevertheless, we could show that both Srt2A- and Srt5M-generated diUbs largely retain their binding affinity towards linkage-specific UBDs, a requirement for triggering diverse cellular signaling events.

In our quest to use Ubl-tools for building distinct Ub chains, we next set out to build a heterotypically linked triUb, using the newly identified orthogonal sortase pair Srt2A/Srt5M in a proof-of-concept experiment. For this, we expressed and purified a bifunctional Ub variant bearing a C-terminal Srt5M motif ('LPT') and GGK at position K48, and incubated it with Ub(LAT) in the presence of Srt2A (Fig. 3a and Supplementary Fig. 5). The obtained K48-linked diUb displaying the LPT motif at its C-terminus was then reacted with Srt5M and Ub-K6GGK, yielding the K48/K6-linked triUb. Importantly, we did not observe any cross-reactivity of Srt2A and Srt5M in the form of unwanted hydrolysis or self-polymerization products, as well as no triUb formation in the absence of Srt5M, confirming the orthogonal behavior of Srt5M towards the 'LAT'-linkage and vice versa. The heterotypically linked triUb was obtained in milligram quantity and its identity was confirmed by liquid-chromatography mass spectrometry (LC-MS, Fig. 3a).

**Building distinct Ub-SUMO2 hybrid chains to investigate their role in DNA damage repair.** Encouraged by successful application of Ubl-tools for creating heterotypic Ub chains, we next tackled the challenge of building biologically relevant hybrid Ub/Ubl chains. Mixed SUMO-Ub chains have recently been discovered to play a major role in DSB repair. At DNA lesion sites, Ub- and SUMO-modifications of chromatin, presumably of H2AX, lead to the recruitment of Rap80, a subunit of the BRCA1-A complex[12,32,33]. Rap80 binds specifically to K63-linked diUbs via its tUIMs (residues 79–96 and 104–124)[30,34], but was also shown to contain an upstream SUMO2-interacting motif (SIM, residues 39–46)[12,32]. This suggests the simultaneous binding of both K63-linked diUb and SUMO2 and the possibility of involvement of Ub-SUMO2 hybrid chains (Fig. 3b). Synthesis of mixed Ub-SUMO2 chains is endogenously facilitated by SUMO-targeted Ub ligases (STUbLs) that specifically recognize and ubiquitylate SUMO chains on substrates[35–37]. For DSB, it has been shown that the human STUbL RNF4 is localized to DNA repair foci and is required for Rap80 and BRCA1-A recruitment, indicating that Ub-SUMO2 hybrid chains may indeed play an important role for DSB repair[38,39]. In fact, studies with heterogeneously linked K63-diUb-SUMO2 hybrid chains generated in vitro by RNF4 indicated a significantly tighter binding of

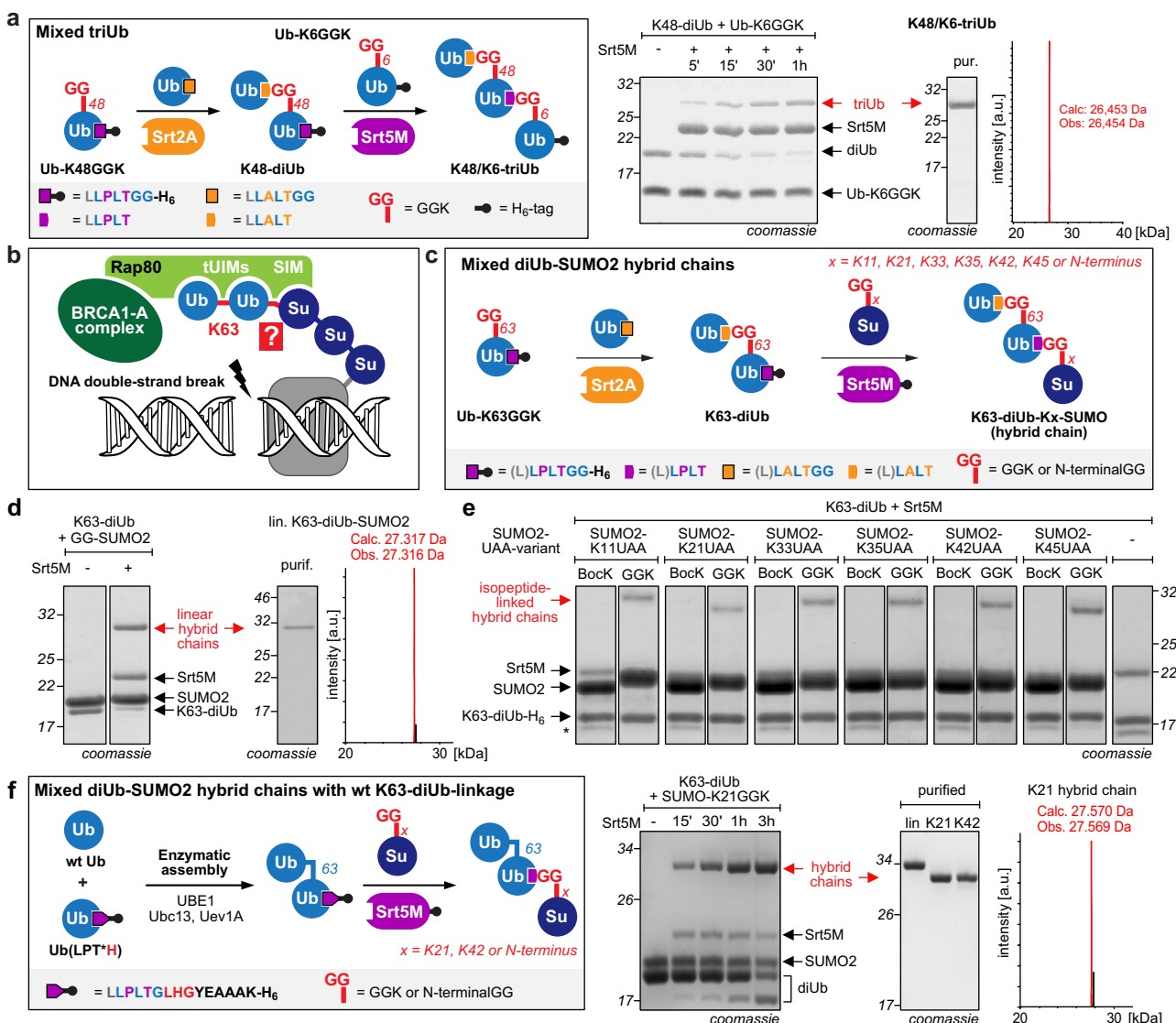

**Fig. 3 Ubl-tools mediated access to heterotypically linked Ubl architectures. a** Left: Schematic representation of the two-step assembly of a heterotypically linked triUb using Srt2A and Srt5M. Right: Generation of K48/K6-linked triUb. SDS-PAGE shows Srt5M-catalyzed transpeptidation between K48-diUb and Ub-K6GGK resulting in the formation of heterotypically K48/K6-linked triUb (K48/K6-triUb). LC-MS analysis confirmed the identity of the K48/K6-triUb. Full gels and densitometric yield determination can be found in Supplementary Fig. 5. **b** Model for Ub-SUMO2 hybrid chain-mediated recruitment of BRCA1-A complex to double strand breaks (DSBs) through SUMO and Ub modifications of chromatin, presumably H2AX[12]. Rap80 contains a SUMO2-interacting motif (SIM) followed by tandem ubiquitin interacting motifs (tUIMs) with high affinity for K63-linked diUb, suggesting the involvement of K63-diUb-SUMO2 hybrid chains. The distinct site of linkage between K63-diUb and SUMO2 is unknown. SUMO2 displays nine possible sites for K63-diUb attachment (the N-terminus and eight lysine residues). **c** Schematic presentation of the two-step strategy to generate linearly or isopeptide-linked K63-diUb-SUMO2 hybrid chains. Ub-K63GGK bearing the Srt5M motif in its C-terminus (PT/LPT) is reacted with Ub(AT/LAT) in the presence of Srt2A. The resulting K63-diUb is subjected to Srt5M-transpeptidation with a SUMO2 variant bearing either a N-terminal GG sequence or a site-specifically introduced GGK moiety. **d** Incubation of Srt2A-generated K63-diUb with a SUMO2 variant bearing a diglycine (GG) moiety at its N-terminus in presence of Srt5M leads to formation of the linear K63-diUb-SUMO2 hybrid chain. LC-MS confirms the integrity of the linearly linked hybrid chain. Densitometric analysis revealed linear hybrid chain formation with 78% yield. **e** SDS-PAGE analysis of Srt5M-mediated transpeptidation of K63-diUb with SUMO2 variants shows specificity for SUMO2-GGK variants. BocK-bearing SUMO2 variants do not yield the respective K63-diUb-SUMO2 hybrid chains. Incubation of K63-diUb and Srt5M in the absence of SUMO2-GGK leads to hydrolysis of the Srt5M recognition motif and hydrolysis of the H₆-tag (depicted by asterisk (*)). UAA stands for unnatural amino acids BocK or GGK. Full gels and densitometrically determined yields as well as LC-MS analysis can be found in Supplementary Fig. 6 and 7a. **f** Left: Schematic presentation of the two-step strategy to generate K63-diUb-SUMO2 hybrid chains combining enzymatic assembly and sortylation. Enzymatic assembly using Ubc13/Uev1A gives access to K63-diUb bearing the LPT*H motif at the C-terminus of the proximal Ub. Srt5M-mediated transpeptidation with a SUMO2 variant bearing either a N-terminal GG sequence or a site-specifically introduced GGK moiety results in formation of diUb-SUMO2 hybrid chains with wt linkage between the Ub moieties. Right: SDS-PAGE analysis of Srt5M-mediated transpeptidation of wt K63-diUb with SUMO2 variants as well as SDS-PAGE and LC-MS analysis of the purified hybrid chains. Full gels and densitometric yield determination can be found in Supplementary Fig. 7c. Consistent results were obtained over at least three replicate experiments. Source data are provided as a Source Data file.

hybrid chains to Rap80 compared to K63-diUb and SUMO2 alone[12]. As RNF4 is capable of catalyzing Ub attachment to the N-terminus and nearly all lysine residues of SUMO2 in in vitro settings[40], the biological role of Ub-SUMO2 hybrid chains and their binding mode to Rap80 remain, however, largely enigmatic. SUMO2 displays nine possible sites to which Ub can be conjugated (the N-terminus and eight lysine residues K5, K7, K11, K21, K33, K35, K42, and K45) and it is not known which linkage leads to most efficient Rap80 binding. The in vitro promiscuity of RNF4 makes it impossible to access homogenous K63-diUb-SUMO2 chains enzymatically. Recent advances in complex solid-phase protein synthesis have guaranteed access to four differently linked K63-diUb-SUMO2 hybrid chains (linked via the N-terminus, K11, K33, or K42 of SUMO2)[41]. This strategy relies however on expert chemistry procedures and requires mutation of cysteine residues within SUMO2. Also, the resulting hybrid chains were not tested for Rap80 binding.

We envisioned that our two-step Ubl-tools procedure will be optimally suited to access adequate amounts of homogenous material of differently linked K63-diUb-SUMO2 hybrid chains in a straightforward manner (Fig. 3c). We expressed and purified bifunctional Ub-K63GGK displaying a Srt5M motif (either PT or LPT) followed by a $H_6$-tag at its C-terminus and incubated it with a Ub variant bearing the Srt2A motif within its C-terminus (Ub(AT) or Ub(LAT)) and Srt2A to generate K63-linked diUb. In parallel, we expressed SUMO2 variants bearing GGK at different lysine positions (K11, K21, K33, K35, K42, or K45) to create isopeptide-linked chains and a SUMO2 variant displaying an N-terminal GG moiety to access a linear K63-diUb-SUMO2 hybrid chain. Incubation of the different SUMO2 variants with K63-diUb displaying the Srt5M motif (PT or LPT) at its proximal Ub C-terminus in the presence of Srt5M led to formation of all seven differently linked hybrid chains (Fig. 3d, e and Supplementary Fig. 6 and 7a). While Srt5M-catalyzed assembly of hybrid chains containing the minimal PT motif between K63-diUb and SUMO2 took several days to reach ~40% yield (based on K63-diUb input), hybrid chains containing the extra leucine spacer within the Srt5M motif linking K63-diUb and SUMO2 yielded up to 65% product formation (based on K63-diUb) within 1 h (Supplementary Fig. 6 and 7a). Importantly, SUMO2 variants bearing the amino acid BocK (Nε-tert-butoxycarbonyl-L-lysine) instead of GGK did not lead to hybrid chain formation in the presence of Srt5M and K63-linked diUb, underlining the specificity of Ubl-tools (Fig. 3e and Supplementary Fig. 6b and 7a). The differently linked K63-diUb-SUMO2 hybrid chains were purified by affinity and size-exclusion chromatography and characterized by LC-MS (Fig. 3d and Supplementary Fig. 6c).

In order to test the binding affinity of K63-diUb-SUMO2 chains towards Rap80, we expressed and purified the N-terminal Rap80 domain harboring the SIM and the tUIMs (amino acids 1–137 with a seven-alanine linker between the tUIMs[30]) and equipped it with an N-terminal $H_6$-tag for PD experiments (Supplementary Fig. 8a). Previous NMR studies of the Rap80-SIM domain in complex with SUMO2 (PDB: 2N9E)[42], as well as of the Rap80-tUIMs bound to K63-diUb (PDB: 2RR9)[34] showed distinct electrostatic and hydrophobic interactions and independent binding of SUMO2 and K63-linked diUb. The SIM and tUIMs are however separated by a ~30 amino acid long unstructured linker, potentially allowing for distinct binding modes of differently linked K63-diUb-SUMO2 chains (Fig. 4a). We performed PD experiments of our hybrid chains with bead-immobilized Rap80$_{(1–137)}$ (Fig. 4b). Densitometric analysis of Coomassie-stained SDS-PAGE gels showed that hybrid chains with a linkage site located within the flexible N-terminus of SUMO2 (via N-terminus or via K11 of SUMO2) showed similar binding affinity towards Rap80. Not surprisingly, K33-, K35-, and

K42-linked hybrid chains showed decreased binding affinity compared to the linear chain. These residues are located in the α1-helix or ß2-strand of SUMO2, which sandwich the ß-strand of the SIM of Rap80 (residues $F_{40}IVI$), forming an intermolecular ß-sheet (Supplementary Fig. 8b). Ubiquitylation of these residues may interfere with SIM binding. This is corroborated by a previous NMR study of SUMO2-SIM that shows that these lysine residues are involved in electrostatic interactions with SIM-residues[42]. For the K45-linked hybrid chain we could observe slightly restored binding affinity more similar to the linear hybrid chain, implying that attachment of K63-diUb at this position may no longer sterically impair SIM binding. Most strikingly, the hybrid chain variant, in which K21 serves as attachment point for K63-diUb showed two-fold increased affinity compared to the linearly linked hybrid chain. K21 is part of the ß1-strand of SUMO2 and points away from the SUMO2-SIM interaction. Ubiquitylation of this residue may therefore not interfere with proper SUMO2 binding. In order to investigate binding affinities and structural features of the binding modes of different hybrid chains towards Rap80, we set out to determine binding constants via fluorescence anisotropy using fluorescently labeled Rap80 and to perform NMR titration experiments using $^{15}N$-labeled Rap80 with three differently linked hybrid chains. To best resemble the functional and structural integrity of endogenous K63-diUb-SUMO2 hybrid chains, we built hybrid chains by combining enzymatic assembly of K63-linked diUb with sortylation using Srt5M (Fig. 3f). For this, we first assembled natively linked K63-diUb bearing a Srt5M motif (LPT) at its proximal Ub-C-terminus using the E1 enzyme UBE1 and the K63-linkage-specific E2-enzymes Ubc13 and Uev1a (Supplementary Fig. 7b). In order to guarantee distinct formation of this diUb variant, we incubated unmodified Ub with an Ub variant bearing a Srt5M recognition motif (LLPLTG) lacking the C-terminal glycine G76, followed by a short linker sequence (LHGYEAAAK). Omitting G76 that is not needed for Srt5M recognition guarantees orthogonality towards E1 and E2 enzymes. The short linker assures good sortase accessibility and contains a masked $Ni^{2+}$-binding peptide (GLHG) that boosts sortylation efficiency by inactivating the emerging nucleophile through $Ni^{2+}$ complex formation (Supplementary Fig. 7c)[43]. Using this approach, we built linear, K21- and K42-linked diUb-SUMO2 hybrid chains containing the wt linker between the two K63-linked Ubs and the LPT linker between K63- diUb and SUMO2 in good yields using K63-diUb and SUMO2-GGK in equimolar ratios (Fig. 3f and Supplementary Fig. 7c, d).

We first determined binding constants of these three hybrid chains using a fluorescently labeled Rap80 construct (Rap80$_{(35–124)}$) using fluorescence anisotropy (Fig. 4c and Supplementary Fig. 8a). In excellent agreement with our PD-data we observed three times tighter Rap80-binding for the K21-linked hybrid chain ($K_D = 0.4\,\mu M$) than for the linearly linked diUb-SUMO2 ($K_D = 1.2\,\mu M$). Correspondingly, the K42-linked chain showed six-fold weaker binding ($K_D = 2.5\,\mu M$) than the K21-linked chain. To characterize the structural features of the different binding modes of these diUb-SUMO2 chains towards Rap80, we expressed and purified uniformly $^{15}N$-labeled Rap80$_{(35–124)}$ (Supplementary Fig. 8a). The secondary structure of Rap80 was analyzed based on $^{13}C_\alpha$ and $^{13}C_\beta$ secondary chemical shifts and reveals that Rap80 shows an unstructured region from amino acid 34–56, shows a helical propensity for residues 62–79, and an α-helical region from amino acid 79–122 (Supplementary Fig. 8c). Titrating the three differently linked unlabeled diUb-SUMO2 chains to this construct, led to differential line broadening, shown by signal intensity reductions I/I(ref), of the backbone amide resonances of the tUIMs and SIM in Rap80, indicating binding of diUb and SUMO2 to these

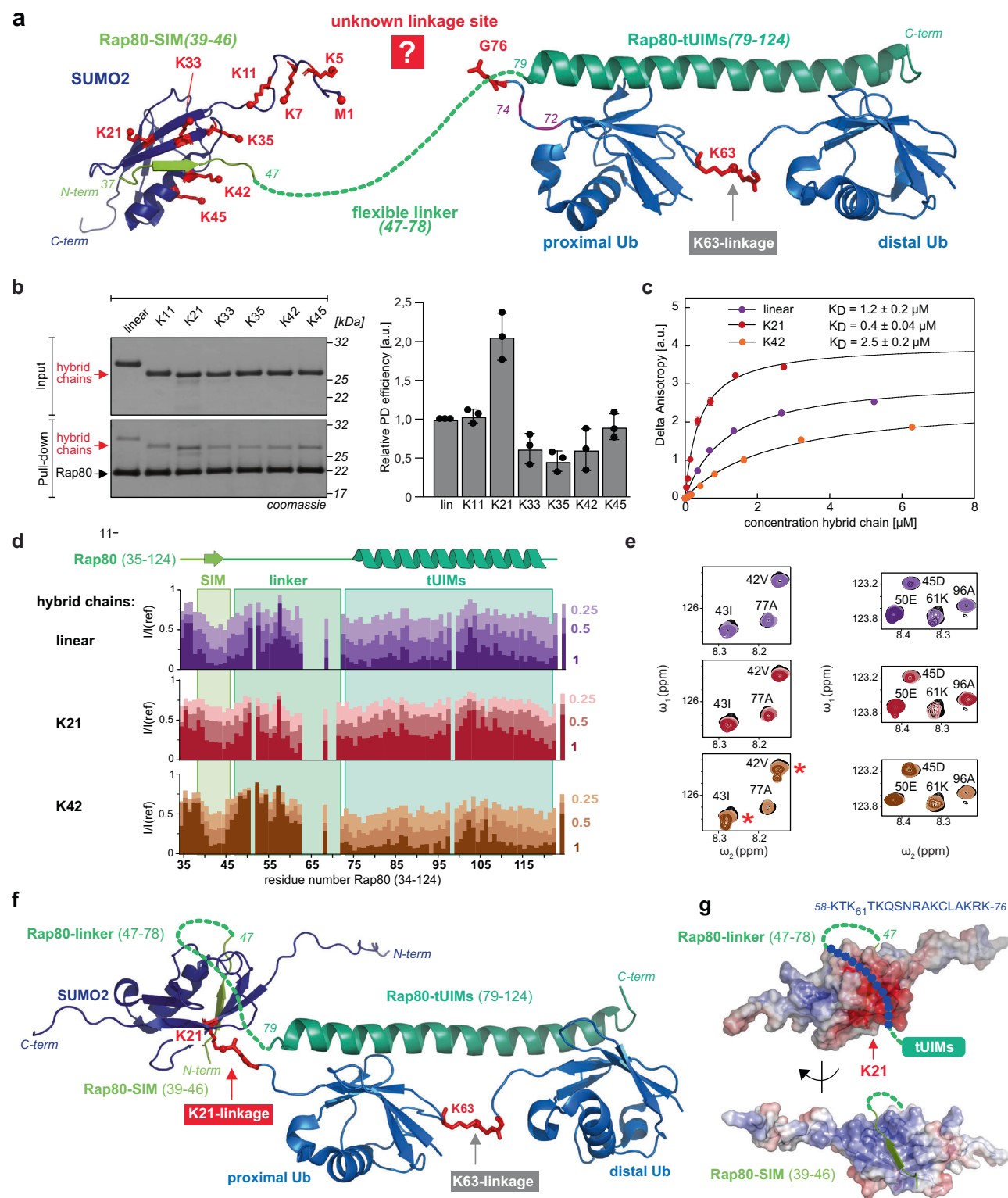

regions, as expected (Fig. 4d). Notably, we also observed line broadening in the unstructured linker region (amino acids 47–62) connecting SIM and tUIMs, suggesting an additional binding interface involving this linker segment (Fig. 4d). Strikingly, this effect was most pronounced for the K21-linked hybrid chain, as observed by stronger signal intensity reductions in the linker region, especially for the positively charged region around lysine K61 (residues 59–75, Fig. 4e). Moreover, the SIM-binding interface is extended and involves also residues around the archetypical hydrophobic ß-strand ($F_{40}IVI$). In contrast, titrations with the K42-linked chain, led to a drop in signal intensity only within the $F_{40}IVI$ core motif, while flanking regions and residues in the subsequent linker (residues 47–62) were largely unaffected. Furthermore, with the K42-linked chain, we observed appearance of a second low populated set of peaks for residues $F_{40}IVI$ within the SIM-ß-strand (Fig. 4e). This is consistent with either parallel or antiparallel ß-strand binding to SUMO as suggested previously (Supplementary Fig. 8d)[42] and corroborates that the K42-linked

**Fig. 4 Characterization of diUb-SUMO2 hybrid chains. a** Structural insights into interactions between Rap80-SIM$_{(39-49)}$ and SUMO2 (PDB: 2N9E[42]) and between Rap80-tUIMs$_{(79-124)}$ and K63-diUb (PDB: 2RR9[34]). SIM and tUIMs are separated by a ~30 aa flexible linker. Possible attachment points of K63-diUb to SUMO2 (M1, K5, K7, K11, K21, K33, K35, K42, and K45) are depicted by red sticks; the distinct linkage site that leads to most efficient Rap80 recruitment is not known. **b** SDS-PAGE analysis of purified hybrid chains and exemplary pull-down analysis. N-terminally H$_6$-tagged Rap80$_{(1-137)}$ was immobilized on beads and incubated with 3.5 μM of hybrid chains and analyzed via Coomassie-stained SDS-PAGE (left). Densitometric analysis of pull-down SDS-PAGE gels (right; average values and error bars (s.d.) were calculated from three different experiments ($n = 3$)). **c** K$_D$ determination via fluorescence anisotropy: Linear, K21- and K42-linked hybrid chains were titrated to fluorescently labeled Rap80$_{(35-124)}$. Delta anisotropy was plotted against hybrid chain concentration and fitted with a single-site binding model to determine K$_D$ values. Average values and error bars (s.d.) were calculated from three different experiments ($n = 3$). All data processing was performed using Kaleidagraph software (Synergy Software, Reading, UK). **d** NMR titration series of linear- (purple), K21- (red), and K42-linked (brown) diUb-SUMO2 hybrid chains to $^{15}$N-labeled Rap80$_{(35-124)}$ (1:0.25, 1:0.5, and 1:1 ratios). Intensity ratios between the titration points and the reference spectrum I/I(ref) are plotted onto the sequence with regions for SIM, linker, and tUIMs highlighted in colored boxes. **e** Zoomed views of $^1$H$^{15}$N-HSQC-spectra overlays show representative residues for the SIM (42 V, 43I, and 45D), the linker region (50E, 61K) and the tUIMs (77A, 96A). Red stars indicate a second set of peaks for titration with K42-linked hybrid chain. **f** Model of interaction between Rap80 and a K63-diUb-K21-SUMO2 hybrid chain. The tUIMs of Rap80 closely bind the Ile44 patches of the K63-linked diUb while the binding between SUMO2 and the SIM of Rap80 occurs in a defined binding groove of SUMO2. In this binding mode, the flexible linker (47–78) will adopt a kinked conformation to allow Rap80 binding of the sterically more compact K21-linked hybrid chain. The scheme was created based on the NMR structure of the tUIMs of Rap80 bound to a K63-linked diUb (PDB: 2RR9) and the NMR structure of SUMO2 bound to a peptide resembling the phosphorylated SIM of the Rap80 N-terminus (PDB: 2N9E). **g** Analysis of surface charge of SUMO2. We propose that the K21-linked diUb-SUMO2 hybrid chain forces the Rap80 linker into a kinked conformation thereby localizing the positively charged linker around K61 (59–75) into a negatively charged region of SUMO2. The scheme is based on PDB: 2N9E[42]. Source data are provided as a Source Data file.

hybrid chain hinders an optimal binding of the SIM and linker region, reflected by its overall lower binding affinity.

Thus, the NMR data and binding affinity measurements suggest a previously unknown binding mode between diUb-SUMO2 hybrid chains and Rap80 that relies on contributions of the ~30 amino acid long linker to enable simultaneous accessibility of SIM and tUIMs. We propose that the K21-linked diUb-SUMO2 hybrid chain forces this linker into a kinked conformation and localizes it near a negatively charged region at the SUMO2 surface adjacent to the hydrophobic SIM-binding groove (Fig. 4f, g). Charge complementarity involving Rap80 linker residues 59–75 and the negatively charged groove on SUMO2 may thus further enhance the Rap80/SUMO2 interaction, consistent with the higher binding affinity compared to the linear and K42-linked hybrid chain (Fig. 4 and Supplementary Fig. 8).

**Ubl-tools allows to charge complex Ub-architectures site-specifically onto target proteins.** Encouraged by the performance and robustness of Ubl-tools to create homogenous material of differently linked hybrid chains, we envisioned that Ubl-tools could be expanded even further. As mentioned above, the functional diversity of ubiquitylation stems from distinct polyUb architectures that are installed site-specifically onto substrate proteins to elicit unique cellular outcomes[3,8]. Strategies controlling for chain type, chain length, and chain position on a substrate protein are therefore essential to understand mechanistic roles of these complex modifications and their impact on critical cellular processes. Arguably, the best studied example is the involvement of polyUb chains on proteasomal protein degradation[5]. The archetypal signal for proteasomal degradation consists in attachment of a K48-linked tetraUb chain to a lysine residue of a target protein[19]. Recent studies have, however, shown that many branched chains elicit particularly efficient degradation[9,10], raising the question what defines the 'perfect' proteasomal degradation signal[3]. Advances in solid-phase peptide synthesis combined with native chemical ligation strategies have recently facilitated the total synthesis of tetraubiquitylated α-globin and have shed light on how Ub topologies may regulate the degradation of proteasomal substrates[44]. As total protein synthesis requires, however, advanced chemistry and is furthermore limited to small, refoldable, and cysteine-free proteins, we envisioned that Ubl-tools may be easily adapted to charging

complex homo- or heterotypic Ubl chains onto target proteins. To this end, we decided to combine enzymatic linkage-specific Ub assembly with iterative sortylation to provide a flexible platform for generating and charging complex Ub chains onto substrate proteins. For site-specifically attaching a K48-linked tetraUb motif onto target proteins, we first set out to enzymatically generate distinctly modified K48-linked diUbs, namely diUb-A and diUb-B, carrying the Srt5M and Srt2A recognition motifs in their C-termini (Fig. 5a). For access to diUb-A, we incubated Ub-K48GGK with a Ub variant bearing the C-terminal Srt5M recognition motif (LPT) in the presence of the E1 enzyme UBE1 and the K48-linkage-specific E2 enzyme CDC34. DiUb-B was created in a similar fashion using wt Ub and a Ub variant bearing a Srt2A-motif-modified C-terminus (LAT). In order to guarantee distinct formation of diUb-A and diUb-B and to prevent uncontrolled polymerization, the starting mono-Ubs were carefully designed and equipped with a minimal sortase recognition motif lacking the C-terminal G76, followed by a rigid linker sequence (YEAAAK). This guarantees orthogonality to the endogenous ubiquitylation machinery, as lack of G76 precludes recognition by endogenous E1 and E2-enzymes (Supplementary Fig. 9a)[18]. Enzymatically generated diUb-A and diUb-B were purified by affinity and size-exclusion chromatography and were obtained in mg quantities (Fig. 5b and Supplementary Fig. 9b, c). Transpeptidation of diUb-A and diUb-B using Srt2A and purification via affinity and size-exclusion chromatography yielded mg-quantities of homogenously K48-linked tetraUb (Fig. 5c and Supplementary Fig. 10a, b). The K48-linked tetraUb contains two wt linkages, as well as a Srt2A-motif linkage (LAT) and bears the Srt5M motif (LPT) at its C-terminus. Incubation with the hydrolase USP2 resulted in the expected pattern, with both LAT and LPT linkages being recalcitrant to DUB cleavage, while wt linkages are completely hydrolyzed by USP2 (Supplementary Fig. 10c). To charge K48-tetraUb site-specifically onto a target protein, we performed a second sortylation reaction using a POI bearing GGK at a user-defined site (here super-folder green fluorescent protein, sfGFP-N150GGK) and Srt5M (Fig. 5d).

We next set out to demonstrate that Ubl-tools is not limited to charging homotypically linked Ub chains to target proteins, but can also be used to site-specifically attach heterotypic branched chains to substrates. Branched Ub polymers are emerging cellular signals and have been shown to play important roles in endocytosis of host proteins by viruses (K11/K63-branched

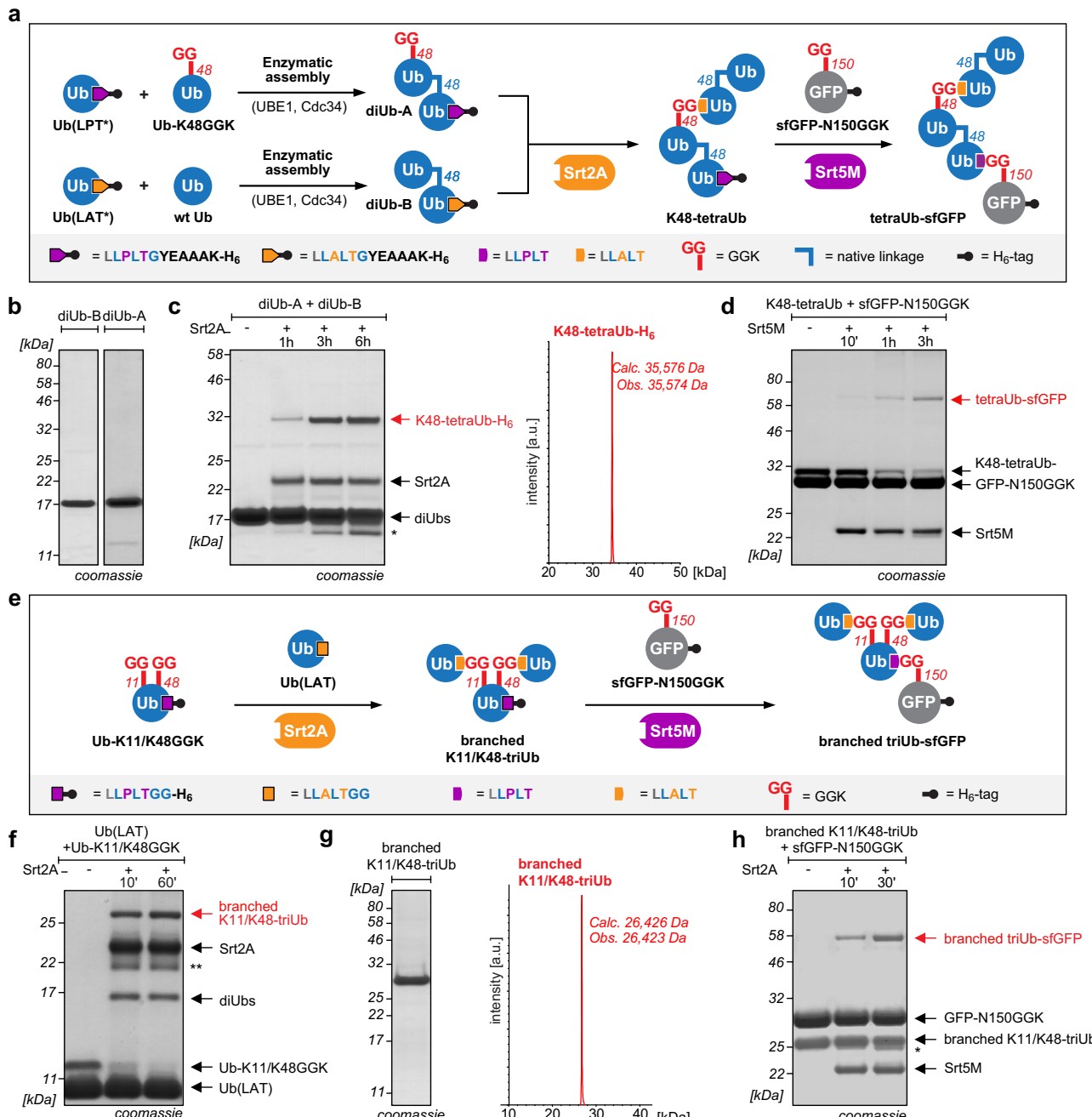

**Fig. 5 Generating defined homo- and heterotypic polyUb-POI conjugates. a** Combining Ubl-tools with enzymatic Ub assembly allows generation of K48-tetraUb, site-specifically charged onto a protein of interest, here sfGFP. **b** SDS-PAGE analysis of enzymatically accessed and distinctly modified K48-linked diUb-A and diUb-B. Full gels as well as LC-MS analysis can be found in Supplementary Fig. 9b, c. **c** Incubation of diUb-A and diUb-B in the presence of Srt2A leads to formation of K48-tetraUb, as depicted by SDS-PAGE analysis and whose integrity is corroborated by LC-MS. Single asterisk (*) denotes hydrolysis of Srt2A recognition motif and cleavage of H₆-tag. Full gels and densitometrically determined yields can be found in Supplementary Fig. 10b. **d** SDS-PAGE analysis showing Srt5M-catalyzed charging of K48-linked tetraUb onto sfGFP-N150GGK. Densitometric analysis revealed formation of tetraUb-sfGFP formation in 45% yield. **e** Schematic representation showing Ubl-tools generated branched Ub chains (K11/K48 branched) and their site-specific attachment to a POI. **f** SDS-PAGE analysis displaying the formation of K11/K48-branched triUb by incubation of Ub-K11/K48GGK with Ub(LAT) and Srt2A. Double asterisks (**) denote an impurity in the Srt2A-stock. Full gels and densitometrically determined yields can be found in Supplementary Fig. 13b. **g** Integrity of purified K11/K48-branched triUb is shown by SDS-PAGE analysis and LC-MS. **h** Incubation of K11/K48-branched triUb with sfGFP-N150GGK in presence of Srt5M leads to the formation of defined branched triUb-sfGFP conjugate. Single asterisk (*) denotes hydrolysis of Srt5M recognition motif and cleavage of H₆-tag. Densitometric analysis revealed branched triUb-sfGFP formation with 33% yield. Consistent results were obtained over at least three replicate experiments. Source data are provided as a Source Data file.

chains)[3] and by enhancing proteasomal degradation of cell cycle proteins (K11/K48-branched chains)[9]. Strategies to generate branched chains rely hitherto on the combination of different linkage-specific E2-enzymes with mutant Ubs and DUBs[17], sophisticated chemical protein synthesis protocols[13,45,46], or on using click chemistry to produce non-natively triazole-linked branched chains[14]. Furthermore, approaches based on thiol-ene coupling in combination with cysteine-mutants[15,47] and silver-catalyzed chemical condensation together with genetic code expansion[48] have been used to build simple branched triUbs.

We envisioned that Ubl-tool should be easily adapted to generate branched chains. For this we site-specifically modified Ub with two GGK moieties (K48/K63, K6/K63, K6/K48, K11/K63, K11/K48, and K6/K11) and incubated these Ub variants with Ub(LAT) in the presence of Srt2A. Gratifyingly, we observed successful formation of branched triUbs for all six double-GGK-bearing Ub variants (Supplementary Fig. 11). In a similar fashion, also enzymatically generated K48-linked diUb-B, bearing the Srt2A recognition motif at the proximal C-terminus (Supplementary Fig. 9c), can be charged twice on double-GGK-modified Ub via Srt2A-transpeptidation, giving rise to symmetrical branched pentaUbs (Supplementary Fig. 12). By equipping the C-terminus of the K11/K48-GGK-modified Ub monomer with the orthogonal Srt5M recognition motif, we furthermore show that also branched Ub chains can be site-specifically attached onto a target protein bearing GGK in a robust and straightforward manner (Fig. 5e–h and Supplementary Fig. 13).

## Discussion

We report Ubl-tools as an easily implementable strategy for the generation of complex Ub/Ubl architectures. Ubl-tools relies on identification of a pair of orthogonal sortase enzymes that can be used in an iterative and modular fashion to grant access to isopeptide-linked, user-defined homotypic, and heterotypic Ub/Ubl chains. We show the straightforward workflow of Ubl-tools by generating mg quantities of seven differently linked diUb-SUMO2 hybrid chains with potential involvement in DNA damage response. Using fluorescence anisotropy and NMR titrations, we investigate the binding preferences of these distinct hybrid chains towards Rap80, a Ub-SUMO chain receptor that mediates and regulates BRAC1-A recruitment to DNA damage sites. This led us to the identification of a previously unknown Rap80-binding mode via a previously uncharacterized diUb-SUMO2 hybrid chain, in which the K63-linked diUb is attached to K21 of SUMO2 via an isopeptide bond.

Apart from SUMO2 also other Ubls, such as Nedd8 and ISG15 have been shown to form Ub/Ubl hybrid chains[11]. In lack of knowledge on enzymes that generate these hybrid chains and in absence of a robust and easy methodology to chemically access some Ubl proteins, their biological roles remain cryptic. Importantly, we have shown before that sortylation is compatible with other Ubls such as for example SUMO[18]. We therefore envision that Ubl-tools can be easily applied to all different Ubls by introducing the Srt2A/Srt5M recognition motifs into their respective C-termini (leading to just one or two point mutations), constituting thereby a valuable tool for generating these otherwise inaccessible Ub/Ubl topologies.

A small adaptation makes Ubl-tools orthogonal to the endogenous ubiquitylation machinery, enabling the combined use of Ubl-tools and enzymatic Ub assembly to build even more complex Ub/Ubl topologies such as branched chains. Strategically combining orthogonal sortylation with enzymatically generated diUbs, enables the generation of complex topologies where we can place DUB-resistant and DUB-susceptible linkages at defined positions. As sortase-catalyzed ubiquitylation works under native

conditions and enables modification of complex, nonrefoldable proteins, Ubl-tools provides a straightforward mean to charge such complex topologies site-specifically onto a user-chosen substrate protein. Exemplarily, we show the site-specific modification of sfGFP with a homotypic K48-linked tetraUb motif as well as with a K11/K48-branched triUb, Ub signals that have been linked with particularly efficient proteasomal degradation[9,19]. Along these lines, we envision that Ubl-tools will become a valuable approach for facilitating structural studies by cryo-electron microscopy (cryo-EM) to decipher how the proteasome engages various polyUb-substrates. In absence of a general and robust protocol for generating endogenous substrate proteins site-specifically modified with distinct Ub chains, previous cryo-EM studies have relied on artificial substrates, enzymatically decorated with Ub chains of random lengths[49,50], failing thereby in providing mechanistic insights into how different Ub chain binding (e.g. K48-linked tetraUb versus K11/K48-branched Ub chains) affects the conformational dynamics of the proteasome. Structural studies of proteasomes, engaged with distinct polyUb-substrates, as obtainable by Ubl-tools, will therefore help in addressing the many challenges in developing a complete mechanistic understanding of substrate processing at the proteasome[51].

Besides providing a general and easily applicable method for accessing defined Ub/Ubl chains and distinct polyUb/Ubl-POI conjugates, Ubl-tools can be easily modified into a tool for identifying receptor and effector proteins that translate these complex Ub/Ubl patterns into a specific outcome for the modified protein, a challenging and relevant endeavor, as crucial details about how e.g. branched or hybrid Ub chains control cellular information flow are currently unknown. In such a scenario, the distal Ub moiety within a distinct Ub/Ubl chain will be equipped with a photocrosslinker or chemical crosslinker moiety in form of a site-specifically incorporated unnatural amino acid and used for proteomic identification of interacting proteins or for chemical stabilization of transient Ub/Ubl chain receptor protein complexes[52,53].

We anticipate that Ubl-tools may be further expanded by discovery and engineering of sortase variants with different and orthogonal recognition motifs[22]. Furthermore, other transpeptidases with minimal recognition motifs such as *Oa*AEP1[54], a recently characterized efficient asparaginyl endopeptidase that is readily produced in bacteria, may prove useful for extending Ubl-tools.

In conclusion we describe a modular and robust tool for accessing defined Ub/Ubl topologies, including hybrid and branched chains, site-specifically conjugated to a user-chosen POI. As Ubl-tools is easily implementable in typical biology labs we envision its potential to provide immediate impact within the Ub-field and spur biological discovery.

## Methods

**Plasmids and reagents.** Human codon optimized SUMO2, Ub, and H6-Rap80(1–137)−7A-Linker were purchased as DNA Strings (GeneArt, Thermo Fisher) and cloned into pPylT and pET17b vectors via standard restriction cloning (see Supplementary Table 3). Point mutations, insertions, and deletions were introduced using Site-directed, Ligase-Independent Mutagenesis (SLIM)[15]. Srt5M, Srt4S, and Srt2A in pET29b vectors were purchased (Addgene plasmids #75144, #75146, and #75145). SENP2 in pET28a was also purchased (Addgene plasmid #16357). Oligonucleotide primers were designed with NEBuilder and purchased from Sigma–Aldrich (see Supplementary Tables 1 and 2). Amino acid sequences of all proteins are listed in Supplementary Note 1.

All solvents and chemical reagents were purchased from Sigma–Aldrich, Carbolution, Acros Organics or Fisher Scientific and were used without further purification unless otherwise stated. Protein LC-MS was carried out on an Agilent Technologies 1260 Infinity LC-MS system with a 6310 Quadrupole spectrometer. The solvent system consisted of 0.1% formic acid in water as solvent A and 0.1% formic acid in ACN as solvent B. Proteins were measured on a Phenomenex Jupiter C4 300 A LC Column (150 ×2 mm, 5 µm). The protein samples were analyzed in positive mode as well as by UV absorbance at 193, 254, and 280 nm. Fifteen percent

SDS-PAGE gels were run (170 V for 60 min) on a Bolt[TM] Mini Gel Tank (Invitrogen) system. Gels were stained with Quick Coomassie Stain (Generon). Color Prestained Protein Standard, Broad Range 11–245 kDa (NEB) or Broad Range 10–250 kDa (NEB) was used as the protein marker. Protein and DNA concentrations were measured on a NanoPhotometer® NP60 (Implen). $H_6$-UBE1 was purchased from Boston Biochem (Cat. No. E-304-050). Ni-NTA agarose was purchased from Jena Bioscience (Cat. No. AC-501). Western blots were carried out on iBlot 2 Dry Blotting System (Life Technologies) using Method P0 (20 V for 1 min, 23 V for 4 min, 25 V for 2 min). After blotting, the nitrocellulose/PVDF membrane was blocked with 5% skim milk powder solution in 1× TBST buffer (1 h, RT) and treated with the desired antibody according to the manufacturer's instructions. Proteins were visualized with WesternBright ECL-spray (Advansta) using an iBright FL1500 imaging system (Thermo Fisher Scientific). Anti-ubiquitin K63-linkage-specific antibody was purchased from Abcam (ab179434, 1:5000) and anti-mouse secondary antibody (Cat. No. A4416, 1:5000) was purchased from Sigma–Aldrich. $N^6$-((2-azidoacetyl)glycyl)-L-lysine (Azido-GGK or AzGGK) was synthesized via solid-phase peptide synthesis (SPPS) as described before[18].

## Protein expression and purification

*Preparation of GGK-bearing POIs.* Chemically competent *E. coli* K12 cells were co-transformed with pPylT_POI (which encodes *Mb* tRNA$_{CUA}$ and the C-terminally $H_6$-tagged POI with one or two TAG codons at the denoted positions (see Supplementary Table 3)) and pBK_aaRS (which encodes either the wt *Mb* aaRS or *Mb* AzGGKRS) plasmids. After recovery with 1 mL of SOC medium for 1 h at 37 °C, the cells were cultured overnight in 50 mL of noninducing medium[55] supplemented with tetracycline (17.5 μg/mL) and ampicillin (100 μg/mL) at 37 °C, 200 rpm. The overnight culture was diluted to an $OD_{600}$ of 0.05 in autoinduction medium[55] containing antibiotics (tetracycline (8.75 μg/mL) and ampicillin (50 μg/mL)) and the corresponding unnatural amino acid (UAA; either 2 mM BocK or 4 mM AzGGK). After overnight-incubation at 37 °C the cells were harvested by centrifugation (4000 × g, 20 min, 4 °C), flash frozen in liquid nitrogen and stored at −80 °C. The obtained cell pellets were thawed on ice and resuspended in lysis buffer (20 mM Tris pH 8.0, 300 mM NaCl, 0.1 mg/mL DNase I (AppliChem), one cOmplete[TM] protease inhibitor tablet (Roche) and 0.175 mg/mL PMSF). Afterwards, the cell suspension was incubated on ice for 30 min and sonicated with cooling in an ice-water bath. The lysed cells were centrifuged (15,000 × g, 40 min, 4 °C) and the clear lysate was added to 1 mL Ni-NTA slurry/1 L culture (Jena Bioscience) equilibrated with wash buffer (20 mM Tris pH 8.0, 300 mM NaCl and 30 mM imidazole). Afterwards, the mixture was incubated with agitation for 1 h at 4 °C. After incubation, the mixture was transferred to an empty plastic column and washed with 10 column volumes (CV) of wash buffer. The protein was eluted in 1 mL fractions with wash buffer supplemented with 300 mM imidazole pH 8.0. The fractions containing the POI (identified via 15% SDS-PAGE) were pooled, concentrated and rebuffered (50 mM Tris pH 7.5 and 150 mM NaCl) using Amicon® centrifugal filter units (Millipore) with a suitable molecular weight cutoff (MWCO). Protein concentration was calculated from the measured $A_{280}$ absorption (extinction coefficients were calculated with ProtParam (https://web.expasy.org/protparam/)). In case of Ub and SUMO the determination of protein concentration using the absorption at 280 nm is inaccurate (due to their low extinction coefficient (ε)), so BCA (Thermo Scientific) and Bradford (Sigma–Aldrich) assays were used to determine the accurate protein concentration. POIs with incorporated UAAs were flash frozen using liquid nitrogen and stored at −80 °C until further use.

Typical expression scale with AzGGK was 1 L. Depending on the position of the TAG codon ~5–20 mg/L SUMO, 5–15 mg/L Ub, and 20–40 mg/L GFP were isolated.

Reduction of the azide moiety of AzGGK to the amine (GGK) was performed on purified proteins by adding 2 eq. of 2-(diphenylphosphino)benzoic acid (2DPBA) or tris-(2-carboxyethyl)-phosphine (TCEP), followed by incubation for 1 h at room temperature and subsequent rebuffering to remove excess of 2DPBA/TCEP. The Staudinger reduction was monitored by LC-MS.

*Cleavage of $H_6$-tag from SUMO2-(KxGGK)-$H_6$ variants using SENP2.* After elution of SUMO2-KxGGK-$H_6$ from Ni-NTA, the eluate (2 mL) was diluted in 20 mL cleavage buffer (50 mM Tris pH 7.5, 150 mM NaCl, and 0.5 mM DTT) and 100 μL of SUMO protease (SENP2) (with a $H_6$-tag, 1.4 mg/mL) were added followed by incubation at RT for 30 min. LC-MS was used to verify completion of the $H_6$-tag cleavage from the C-terminus of SUMO2-KxGGK-$H_6$. After successful cleavage, the reaction mixture was added to equilibrated Ni-NTA (200 μL/1 mg SUMO2-KxGGK) and incubated at 4 °C for 1 h with agitation. After incubation, the mixture was transferred to an empty plastic column, the flow through was collected, concentrated and rebuffered (50 mM Tris pH 7.5, 150 mM NaCl) using Amicon® centrifugal filter units with a 3 kDa MWCO. Protein concentration was determined using BCA (Thermo Scientific™) and Bradford (Sigma–Aldrich) assays. Tagless SUMO2-KxGGK was flash frozen using liquid nitrogen and stored at −80 °C until further use.

*Cleavage of $H_6$-tag from Ub-(KxGGK)-$H_6$ variants using USP2.* $H_6$-tag cleavage from Ub-KxGGK-$H_6$ was performed in an analogous manner as described above

for $H_6$-tag cleavage from SUMO2-KxGGK-$H_6$ using a deubiquitylase (100 μL USP2, 2 mg/mL) instead of SUMO protease (SENP2).

*Expression and purification of sortase mutants.* Chemically competent *E. coli* BL21(DE3) were transformed with pET29b_Srt-$H_6$ or pET29b_Srt-TEV-$H_6$ plasmid (see Supplementary Table 3). After recovery with 1 mL of SOC medium for 1 h at 37 °C, the cells were cultured overnight in 50 mL of 2× YT medium containing kanamycin (50 μg/mL) at 37 °C, 200 rpm. The overnight culture was diluted to an $OD_{600}$ of 0.05 in 3 L of fresh 2× YT medium supplemented with kanamycin (25 μg/mL) and cultured at 37 °C while shaking (200 rpm) until an $OD_{600} = 0.5$–0.8 was reached. IPTG was added to a final concentration of 0.4 mM and protein expression was induced for three hours at 30 °C. The cells were harvested by centrifugation (4,000 × g, 20 min, 4 °C) and resuspended in lysis buffer (50 mM Tris pH 8.0, 300 mM NaCl supplemented with 1 mM MgCl$_2$, 0.1 mg/mL DNase I (AppliChem), one cOmplete[TM] protease inhibitor tablet (Roche) and 0.175 mg/mL PMSF). Cells were lysed by sonication with cooling in an ice-water bath and centrifuged (15,000 × g, 40 min, 4 °C). Afterwards, the cleared lysate was added to Ni-NTA slurry (Jena Bioscience) (1 mL of slurry per 1 L culture) and equilibrated with wash buffer (20 mM Tris pH 8.0, 300 mM NaCl, 30 mM imidazole). Subsequently, the mixture was incubated with agitation for one hour at 4 °C. After incubation, the mixture was transferred to an empty plastic column and washed with 10 CV of wash buffer. The protein was eluted in 1 mL fractions with wash buffer supplemented with 300 mM imidazole pH 8.0. The fractions containing the protein were pooled, concentrated and rebuffered (20 mM Tris pH 7.5, 150 mM NaCl, 5 mM CaCl$_2$) using Amicon® Ultra-4 10 K MWCO centrifugal filter units (Millipore). Enzyme concentration was calculated from the measured $A_{280}$ absorption (extinction coefficients were calculated with ProtParam (https://web.expasy.org/protparam/)). All sortase variants were flash frozen using liquid nitrogen and stored at −80 °C until further use.

Sortase variants harbouring a C-terminal TEV-$H_6$-tag were expressed and purified identically. In order to cleave off the $H_6$-tag, the fractions containing the protein were pooled and 200 μL of TEV protease (1.8 mg/mL) were added. The mixture was transferred to a dialysis tubing (Roth) and the dialysis bag was immersed in 2 L of cold dialysis buffer (25 mM Tris pH 8.0, 150 mM NaCl, 0.5 mM DTT) and stirred at 4 °C overnight. The protein mixture was recovered from the dialysis tubing and centrifuged (15,000 × g, 10 min, 4 °C) in order to precipitate TEV protease. 1 mL of Ni-NTA slurry (Jena Bioscience) per 1 L culture, equilibrated with dialysis buffer, was added to the supernatant and the mixture was incubated with agitation for 1 h at 4 °C. The mixture was then poured into an empty plastic column and the flow through was collected. The Ni-NTA beads were washed twice with 15 mL of wash buffer (20 mM Tris pH 8.0, 150 mM NaCl, 5 mM CaCl$_2$). Flow through and wash fractions containing purified sortase without $H_6$-tag were pooled, concentrated, and rebuffered (50 mM Tris pH 7.5, 150 mM NaCl, 5 mM CaCl$_2$) using Amicon® Ultra-4 10 K MWCO centrifugal filter units (Millipore). Enzyme concentration was calculated from the measured $A_{280}$ absorption (extinction coefficients were calculated with ProtParam (https://web.expasy.org/protparam/)). All sortase variants were flash frozen using liquid nitrogen and stored at −80 °C until further use.

*Expression and purification of tagless ubiquitin and ubiquitin variants.* Chemically competent *E. coli* Rosetta2 (DE3) were transformed with pET17b-ubiquitin plasmid (see Supplementary Table 3). After recovery with 1 mL of SOC medium for one hour at 37 °C, the cells were cultured overnight in 50 mL of 2× YT medium containing ampicillin (100 μg/mL) and chloramphenicol (50 μg/mL) at 37 °C, 200 rpm. The overnight culture was diluted to an $OD_{600}$ of 0.05 in 3 L of fresh 2× YT medium supplemented with ampicillin (50 μg/mL) and chloramphenicol (25 μg/mL) and cultured at 37 °C with shaking (200 rpm) until $OD_{600} = 0.8$–1.0. IPTG was added to a final concentration of 1 mM and protein expression was induced for 4 h at 37 °C. The cells were harvested by centrifugation (4000 × g, 20 min, 4 °C) and resuspended in lysis buffer (50 mM Tris pH 7.6, 10 mM MgCl$_2$, 1 mM EDTA, 0.1% NP-40, 0.1 mg/mL DNAse I, one cOmplete[TM] protease inhibitor tablet and 0.175 mg/mL PMSF). Cells were lysed by sonication with cooling in an ice-water bath and centrifuged (15,000 × g, 40 min, 4 °C).

The cleared lysate was transferred into a glass beaker in an ice-water bath placed on a magnetic stirrer. Precipitation was performed with 35 % perchloric acid until pH 4.0–4.5 was reached. After 5 minutes incubation at 4 °C while stirring, the milky solution was centrifuged (15,000 × g, 40 min, 4 °C) and the supernatant was transferred into a dialysis tubing with a MWCO of 2 kDa (Roth). Dialysis was performed overnight at 4 °C with 50 mM ammonium acetate buffer pH 4.5. The dialysed solution was centrifuged (15,000 × g, 40 min, 4 °C), filtered and purified via a HiTrap SP FF 5 mL cation exchange chromatography (GE, gradient 0–1 M NaCl). Fractions that showed > 95% purity, as judged by SDS-PAGE, were pooled and rebuffered (50 mM Tris pH 8.0, 150 mM NaCl, 5 mM CaCl$_2$) using Amicon® Ultra-15 3 kDa MWCO centrifugal filter units (Millipore). Protein concentration was calculated from the measured $A_{280}$ absorption (extinction coefficients were calculated with ProtParam (https://web.expasy.org/protparam/)). In case of Ub the determination of protein concentration using the absorption at 280 nm is inaccurate (due to the low extinction coefficient (ε)), and therefore BCA (Thermo Scientific™) and Bradford (Sigma–Aldrich) assays were used for accurate protein

concentration determination. All Ub variants were flash frozen using liquid nitrogen and stored at −80 °C until further use.

*Expression and purification of H$_6$-tagged Ub and SUMO variants.* Chemically competent *E. coli* Rosetta2 (DE3) were transformed with H$_6$-tagged Ub/SUMO variants in a pET17b plasmid (see Supplementary Table 3). After recovery with 1 mL of SOC medium for 1 h at 37 °C, the cells were cultured overnight in 50 mL of 2× YT medium containing ampicillin (100 µg/mL) and chloramphenicol (50 µg/mL) at 37 °C, 200 rpm. The overnight culture was diluted to an OD$_{600}$ of 0.05 in 3 L of fresh 2× YT medium supplemented with ampicillin (50 µg/mL) and chloramphenicol (25 µg/mL) and cultured at 37 °C, 200 rpm, until OD$_{600}$ = 0.8–1.0. IPTG was added to a final concentration of 1 mM and protein expression was induced for 4 h at 37 °C. The cells were harvested by centrifugation (4000 × g, 20 min, 4 °C) and resuspended in 20 mL of lysis buffer/1 L of culture (20 mM Tris pH 8.0, 300 mM NaCl, 30 mM imidazole, 0.175 mg/mL PMSF, 0.1 mg/mL DNase I, and one cOmplete$^{TM}$ protease inhibitor tablet (Roche)). The cell suspension was incubated on ice for 30 min and sonicated with cooling in an ice-water bath. The lysed cells were centrifuged (15,000 × g, 40 min, 4 °C), the cleared lysate added to Ni-NTA slurry (Jena Bioscience) (1 mL of slurry per 1 L of culture) and the mixture was incubated with agitation for 1 h at 4 °C. After incubation, the mixture was transferred to an empty plastic column and washed with 10 CV of wash buffer (20 mM Tris pH 8.0, 300 mM NaCl, 30 mM imidazole). The protein was eluted in 1 mL fractions with wash buffer supplemented with 300 mM imidazole pH 8.0. The fractions containing the protein were pooled, concentrated and rebuffered (50 mM Tris pH 7.5, 150 mM NaCl, 5 mM CaCl$_2$) using Amicon® centrifugal filter units with a 3 kDa MWCO (Millipore). Purified proteins were analyzed by 15% SDS-PAGE and mass spectrometry. In case of Ub and SUMO the determination of protein concentration using the absorption at 280 nm is inaccurate (due to their low extinction coefficient, so BCA (Thermo Scientific™) and Bradford (Sigma–Aldrich) assays were used for accurate protein concentration determination. H$_6$-tagged Ub and SUMO variants were flash frozen using liquid nitrogen and stored at −80 °C until further use.

*Expression and purification of H$_6$-Rap80 variants[4].* Chemically competent *E. coli* Rosetta2 (DE3) were transformed with pET28a-H$_6$-Rap80$_{(1–137)}$-7A-Linker (for pull-down experiments), pET28a-H$_6$-Rap80$_{(35–124)}$-S18C-C70S-C121S-7A-Linker (for fluorescence anisotropy experiments with hybrid chains) or pET28a-H$_6$-Rap80$_{(79–124)}$-7A-Linker (for fluorescence anisotropy experiments with K63-linked diUbs) plasmid (see Supplementary Table 3). After recovery with 1 mL of SOC medium for one hour at 37 °C, the cells were cultured overnight in 50 mL of 2× YT medium containing kanamycin (50 µg/mL) and chloramphenicol (50 µg/mL) at 37 °C, 200 rpm. The overnight culture was diluted to an OD$_{600}$ of 0.05 in 500 mL of fresh 2× YT medium supplemented with kanamycin (25 µg/mL) and chloramphenicol (25 µg/mL) and cultured at 37 °C with shaking (200 rpm) until OD$_{600}$ = 0.8. IPTG was added to a final concentration of 0.5 mM and protein expression was induced for 16 h at 18 °C. The cells were harvested by centrifugation (4000 × g, 20 min, 4 °C) and resuspended in lysis buffer (20 mM Tris pH 8.0, 300 mM NaCl, 30 mM imidazole, 1 mM TCEP, 0.175 mg/mL PMSF, 0.1 mg/mL DNase I, and one cOmplete$^{TM}$ protease inhibitor tablet (Roche)). The cell suspension was incubated on ice for 30 min and sonicated with cooling in an ice-water bath. The lysed cells were centrifuged (15,000 × g, 40 min, 4 °C) and the cleared lysate was added to Ni-NTA slurry (Jena Bioscience) (1 mL of slurry per 1 L of culture) equilibrated with lysis buffer. Subsequently, the mixture was incubated with agitation for 1 h at 4 °C. After incubation, the mixture was transferred to an empty plastic column and washed with 10 CV wash buffer (50 mM Tris pH 8.0, 30 mM imidazole pH 8.0, 300 mM NaCl). The protein was eluted in 1 mL fractions with elution buffer (wash buffer supplemented with 300 mM imidazole pH 8.0). The fractions containing H$_6$-Rap80 variants were pooled and concentrated using Amicon® centrifugal filter unit with a 3 kDa MWCO (Millipore) followed by size-exclusion chromatography (SEC) using a Superdex S75 10/300 (GE Healthcare) with SEC buffer (50 mM Tris pH 7.5, 150 mM NaCl). Fractions containing H$_6$-Rap80 variants were pooled and concentrated with Amicon® 3 kDa MWCO centrifugal filter units (Millipore). Protein concentration was determined using BCA (Thermo Scientific™) and Bradford (Sigma–Aldrich) assays (since H$_6$-Rap80 variants do not contain aromatic amino acids with absorbance at 280 nm). H$_6$-Rap80 variants were flash frozen using liquid nitrogen and stored at −80 °C until further use.

*Fluorescence labelling of H$_6$-Rap80$_{(79–124)}$-7A-Linker and H$_6$-Rap80$_{(1–137)}$-S18C-C70S-C121S-7A-Linker.* 20 µM H$_6$-Rap80$_{(79–124)}$-7A-Linker/H$_6$-Rap80$_{(1–137)}$-S18C-C70S-C121S-7A-Linker were incubated with 100 µM Atto488 maleimide (Atto-TEC, Cat. No. AD 488) in PBS at 4 °C. Labelling reaction was monitored via LC-MS. After ~30 min quantitative labelling was achieved and the excess of fluorophor was quenched with 1 mM DTT (10 min, 4 °C). Afterwards the excess of quenched Fluorophor was removed by buffer exchange to PBS using Amicon® 10 kDa MWCO centrifugal filter units (Millipore). Concentration of Atto488-labelled Rap80 was determined via Atto488 absorbance (500 nm, e$_{max}$ = 9.0 × 10$^4$ M$^{−1}$ cm$^{−1}$). Labelled H$_6$-Rap80 variants were flash frozen using liquid nitrogen and stored at −80 °C until further use.

*Expression and purification of isotope labelled H$_6$-Rap80$_{(35–124)}$-7A-Linker.* Uniformly $^{13}$C and/or $^{15}$N-labelled proteins were expressed in M9 minimal medium

supplemented with 2 g/L hydrated [U-$^{13}$C] glucose and/or 1 g/L $^{15}$N NH$_4$Cl and as the sole sources of carbon and nitrogen, respectively. Expression and purification were performed analogously to the above-described procedure. After Ni-NTA purification isotope labelled H$_6$-Rap80$_{(35–124)}$-7A-Linker was applied to size-exclusion chromatography (SEC) using a Superdex S75 10/300 (GE Healthcare) with NMR buffer (20 mM Tris pH 6.8, 150 mM NaCl). Isotope labelled H$_6$-Rap80$_{(35–124)}$-7A-Linker was flash frozen using liquid nitrogen and stored at −80 °C until further use.

*Expression and purification of E2-enzymes (GST-Cdc34, GST-Ubc13, and GST-Uev1A).* Chemically competent *E. coli* Rosetta2 (DE3) were transformed with pGEX-6P-1-UBE2R1 (encoding for GST-Cdc34), pGEX6P1-UBE2N (encoding for GST-Ubc13), or pGEX6P1-UBE2V1 (encoding for GST-Uev1A). After recovery with 1 mL of SOC medium for 1 h at 37 °C, the cells were cultured overnight in 50 mL of 2× YT medium containing ampicillin (100 µg/mL) and chloramphenicol (50 µg/mL) at 37 °C, 200 rpm. The overnight culture was diluted to an OD$_{600}$ of 0.05 in 1 L of fresh 2× YT medium supplemented with ampicillin (50 µg/mL) and chloramphenicol (25 µg/mL) and cultured at 37 °C with shaking (200 rpm) until OD$_{600}$ = 0.8. IPTG was added to a final concentration of 0.25 mM and protein expression was induced for 18 h at 20 °C. The cells were harvested by centrifugation (4000 × g, 20 min, 4 °C) and resuspended in lysis buffer (50 mM Tris pH 8.0, 300 mM sucrose, 50 mM NaF, 2 mM DTT, 0.1 mg/mL DNase I, and one cOmplete$^{TM}$ protease inhibitor tablet (Roche)). The cell suspension was incubated on ice for 30 min and sonicated with cooling in an ice-water bath. The lysed cells were centrifuged (15,000 × g, 40 min, 4 °C), the cleared lysate added to Glutathione Sepharose 4B (GE Healthcare) (0.1 mL of slurry per 100 mL of culture) and the mixture was incubated with agitation for 1 h at 4 °C. After incubation, the mixture was transferred to an empty plastic column and washed with 10 CV of wash buffer (25 mM Tris pH 8.5, 400 mM NaCl, 5 mM DTT). The protein was eluted in 1 mL fractions with elution buffer (wash buffer supplemented with 10 mM GSH pH 8.0). The fractions containing GST-tagged E2-enzymes were pooled and concentrated with Amicon® centrifugal filter units with a 10 kDa MWCO (Millipore) followed SEC using a Superdex S75 16/600 (GE Healthcare) with SEC buffer (50 mM Tris pH 7.5, 150 mM NaCl, 1 mM DTT). Fractions containing the GST-tagged E2-enzymes were pooled and concentrated with Amicon® centrifugal filter units with an appropriate MWCO (Millipore). Protein concentration was calculated from the measured A$_{280}$ absorption (extinction coefficients were calculated with ProtParam (https://web.expasy.org/protparam/)). GST-tagged E2-enzymes were flash frozen using liquid nitrogen and stored at −80 °C until further use.

Facultative GST-cleavage was performed by diluting GST-tagged E2-enzymes to 50 µM in SEC buffer (50 mM Tris pH 7.5, 150 mM NaCl, 1 mM DTT) followed by addition of PreScission Protease (Sigma–Aldrich, Cat. No. GE27-0843-01). After incubation at 4 °C for 1 h preequilibrated Glutathione Sepharose 4B (GE Healthcare) was added to remove GST and PreScission protease. The flow through was collected and applied to SEC using a Superdex S75 16/600 (GE Healthcare) with SEC buffer. Fractions containing the untagged E2-enzymes were pooled and concentrated with Amicon® centrifugal filter units with an appropriate MWCO (Millipore). Protein concentration was calculated from the measured A$_{280}$ absorption (extinction coefficients were calculated with ProtParam (https://web.expasy.org/protparam/)). E2-enzymes were flash frozen using liquid nitrogen and stored at −80 °C until further use.

*Expression and purification of USP2, UCHL3, and SENP2.* USP2[9], UCHL3[10], and SENP2[11] were expressed and purified as previously described.

## Identification of orthogonal sortases via diUb hydrolysis assays

*Preparation of diUbs using sortase.* GGK-bearing acceptor Ub (either with a native C-terminus (e.g. Ub-K6GGK) or with a C-terminal H$_6$-tag (e.g. Ub-K6GGK-H$_6$) or with a C-terminus compatible for a subsequent reaction with an orthogonal sortase (e.g. Ub-K6GGK-(PT)-H$_6$)) was diluted to 20 µM in sortase buffer (50 mM Tris pH 7.5, 150 mM NaCl, 5 mM CaCl$_2$). Afterwards, 100 µM of the donor Ub (displaying a sortase recognition motif at its C-terminus) was added, which was followed by the addition of 20 µM of the corresponding Srt mutant (without H$_6$-tag). Incubation was performed at 37 °C for 1 h when a donor Ub variant with spacer was used and 18 h when a donor Ub variant without spacer was used. Sortase-mediated transpeptidation was stopped by the addition of 200 µM phenyl vinyl sulfone and further incubation for 10 min at 37 °C. Afterwards, affinity purification using Ni-NTA slurry (Jena Bioscience) was performed as described above. Fractions containing a mixture of sortase-generated diUb and unreacted acceptor Ub were pooled and concentrated using Amicon® centrifugal filter units with a 3 kDa MWCO (Millipore). To remove unreacted acceptor Ub, SEC was performed using a Superdex S75 16/600 (GE Healthcare) with 50 mM Tris pH 7.5, 150 mM NaCl. Fractions containing the sortase-generated diUb were pooled and concentrated using Amicon® centrifugal filter units with a 10 kDa MWCO (Millipore). DiUbs were flash frozen using liquid nitrogen and stored at −80 °C until further use.

This protocol was used: (1) to generate all K6-linked diUbs to investigate sortase orthogonality in sortase hydrolysis assays, (2) to gain access to K48- and K63-linked diUbs for the characterization of the PT/LPT/AT/LAT substitutions, (3) to assemble the K48-linked diUb used for the formation of heterotypically linked triUb, and (4) to prepare K63-linked diUbs which serve as a basis for all hybrid chains.

*DiUb hydrolysis assays.* K6-linked diUbs displaying different sortase motifs at their linkage site (LAT, LPT, and LPS) were diluted into sortase buffer to 40 µM followed by addition of 10 µM of different sortase mutants. Reaction mixtures were incubated at 37 °C and 6 µL samples were taken at the denoted time points and quenched by the addition of 4× SDS loading buffer. After boiling at 95 °C for 10 min and centrifugation (14,000 × $g$, 5 min) samples were loaded on 15% SDS-PAGE gels and visualized by Coomassie staining.

### Preparation of natively linked diUbs

*Preparation of natively linked K48-diUb variants.* Assembly of K48-linked diUbs was carried out as previously described with slight adjustments[12]. In short, the assembly reactions contained 50 nM H$_6$-UBE1 (Boston Biochem, Cat. No. E-304-050), 4.5 µM GST-Cdc34 and variable concentrations of Ub mutants (as depicted in Supplementary Fig. 9). Reactions were incubated at 37 °C for typically 16 h in diUb reaction buffer (50 mM Tris pH 7.5, 10 mM MgCl$_2$, 0.6 mM DTT, 10 mM ATP). After incubation, diluted in wash buffer to reduce the overall DTT concentration and 250 µL (per mL reaction volume) of Ni-NTA slurry (Jena Bioscience, equilibrated with wash buffer (50 mM Tris pH 7.5, 150 mM NaCl, 30 mM imidazole)) was added to the reaction and the mixture was incubated agitating for 1 h at 4 °C. After incubation, the mixture was transferred to an empty plastic column and washed with 40 CV of wash buffer to remove GST-Cdc34, unreacted wt Ub, wt K48-diUb, and Ub-K48GGK. H$_6$-tagged proteins were eluted in 0.2 mL fractions with wash buffer supplemented with 300 mM imidazole pH 8.0. The fractions containing the mixture of desired K48-linked diUb (diUb-A or diUb-B) and remaining H$_6$-tagged monoUbs (Ub(LAT*)-H$_6$ and Ub(LPT*)-H$_6$) were pooled and concentrated using Amicon® centrifugal filter units with a 10 kDa MWCO (Millipore). In order to remove remaining H$_6$-tagged monoUbs (Ub(LAT*)-H$_6$ and Ub(LPT*)-H$_6$), SEC was performed using a Superdex S75 16/600 (GE Healthcare) with SEC Buffer (50 mM Tris pH 7.5, 150 mM NaCl). Fractions containing the desired K48-linked diUbs (diUb-A or diUb-B) were pooled and concentrated using Amicon® centrifugal filter units with a 3 kDa MWCO (Millipore). Protein concentration was determined using BCA (Thermo Scientific™) and Bradford (Sigma–Aldrich) assays. DiUb-A and diUb-B were flash frozen using liquid nitrogen and stored at -80 °C until further use.

*Preparation of natively linked K63-diUb variants.* Assembly of K63-linked diUbs was carried out as previously described with slight adjustments[13]. In short, the assembly reactions contained 50 nM H$_6$-UBE1 (Boston Biochem, Cat. No. E-304-050), 2.5 µM Ubc13, 2,5 µM Uev1A, 600 µM Ub(LPT*H)-H$_6$, and 400 µM Ub wt. Reactions were incubated at 37 °C for typically 2 h in diUb reaction buffer (50 mM Tris pH 7.5, 10 mM MgCl$_2$, 0.6 mM DTT, 10 mM ATP). After incubation, the reaction mixture was diluted in wash buffer to reduce the overall DTT concentration and 250 µL (per mL reaction volume) of Ni-NTA slurry (Jena Bioscience, equilibrated with wash buffer (50 mM Tris pH 7.5, 150 mM NaCl, 30 mM imidazole) was added and the mixture was incubated agitating for 1 h at 4 °C. After incubation, the mixture was transferred to an empty plastic column and washed with 40 CV of wash buffer to remove Ubc13, Uev1a, unreacted Ub wt, and wt K63-diUb. H$_6$-tagged proteins were eluted in 0.2 mL fractions with wash buffer supplemented with 300 mM imidazole pH 8.0. The fractions containing the mixture of desired K63-linked diUb and remaining H$_6$-tagged monoUb were pooled and concentrated using Amicon® centrifugal filter units with a 10 kDa MWCO (Millipore). To remove remaining H$_6$-tagged monoUb, SEC was performed using a Superdex S75 16/600 (GE Healthcare) with SEC Buffer (50 mM Tris pH 7.5, 150 mM NaCl). Fractions containing the desired K63-linked diUb were pooled and concentrated using Amicon® centrifugal filter units with a 10 kDa MWCO (Millipore). Protein concentration was determined using BCA (Thermo Scientific™) and Bradford (Sigma–Aldrich) assays. K63-diUb was flash frozen using liquid nitrogen and stored at −80 °C until further use.

### Generation of complex Ub/Ubl topologies using sortase

*Preparation of K48/K6-linked triUb using sortase.* The sortase reaction to generate the triUb was carried out as described above for diUbs. In short, 20 µM of Ub(LAT)-K48-Ub(LPT)-H$_6$ (see Supplementary Fig. 5) was used as donor Ub and 100 µM Ub-K6GGK-H$_6$ were used as acceptor Ub. Srt5M (20 µM, without H$_6$-tag) was added, followed by incubation at 37 °C for 1 h. Affinity purification followed by SEC (50 mM Tris pH 7.5, 150 mM NaCl) to purify the triUb. TriUb was flash frozen using liquid nitrogen and stored at −80 °C. LC-MS analysis was used to verify the identity of the triUb. Analytical assays of triUb formation were performed analogously. Six microliters samples were taken at the denoted time points and quenched by the addition of 4× SDS loading buffer and boiling at 95 °C for 10 min. Samples were loaded on 15% SDS-PAGE gels and visualized by Coomassie staining.

*Preparation of AT/PT- and LAT/LPT-linked hybrid chains using Srt2A and Srt5M.* The sortase reaction to generate AT/PT- and LAT/LPT-linked hybrid chains was carried out similarly as described above for diUbs. In short, 20 µM of Ub(AT)-K63-Ub(PT)-H$_6$/ Ub(LAT)-K63-Ub(LPT)-H$_6$ (see Supplementary Fig. 6 and 7) was used as donor and 100 µM SUMO2-KxGGK (without C-terminal H$_6$-tag, which was removed using SENP2) was used as acceptor. Srt5M (20 µM, with H$_6$-tag) was

added followed by incubation at 37 °C. For the purification of AT/PT-linked hybrid chains, after 72 h at 37 °C Ni-NTA was used to remove unreacted diUb and Srt5M. The concentrated Ni-NTA flow through, containing the desired hybrid chain and the excess of SUMO2-KxGGK, was applied to SEC (50 mM Tris pH 7.5, 150 mM NaCl) for separation. The fractions containing pure hybrid chains (identified by 15% SDS-PAGE) were pooled and concentrated using Amicon® centrifugal filter units with a 10 kDa MWCO. Protein concentration was determined using BCA (Thermo Scientific™) and Bradford (Sigma–Aldrich) assays. Hybrid chains were flash frozen using liquid nitrogen and stored at −80 °C until further use. LC-MS analysis was used to verify the identity of hybrid chains.

Analytical assays for hybrid chain formation were performed analogously. Six microliters samples were taken at the denoted time points and quenched by the addition of 4× SDS loading buffer and boiling at 95 °C for 10 min. Samples were loaded on 15% SDS-PAGE gels and visualized by Coomassie staining.

*Preparation of wt/LPT-linked hybrid chains using enzymatic assembly and Srt5M.* For the preparation of wt/LPT-linked hybrid chains 50 µM of natively linked K63-diUb(LPT*H)-H$_6$ was incubated with 50 µM SUMO2-KxGGK (without C-terminal H$_6$-tag, which was removed using SENP2) in sortase reaction buffer supplemented with 500 µM NiSO$_4$ (see Supplementary Fig. 7). Srt5M (10 µM, with H$_6$-tag) was added followed by incubation at 37 °C. After incubation for 1 h, Ni-NTA was used to remove unreacted diUb and Srt5M. The concentrated Ni-NTA flow through, containing the desired hybrid chain, diUb without C-terminal H$_6$-tag and SUMO2-KxGGK, was applied to SEC (50 mM Tris pH 7.5, 150 mM NaCl) for separation. The fractions containing pure hybrid chains (identified by 15% SDS-PAGE) were pooled and concentrated using Amicon® centrifugal filter units with a 10 kDa MWCO. Protein concentration was determined using BCA (Thermo Scientific™) and Bradford (Sigma–Aldrich) assays. Hybrid chains were flash frozen using liquid nitrogen and stored at −80 °C until further use. LC-MS analysis was used to verify the identity of hybrid chains.

Analytical assays to investigate the impact of selective nucleophile quenching facilitated by NiSO$_4$ were performed analogously. Six microliters samples were taken at the denoted time points and quenched by the addition of 4× SDS loading buffer and boiling at 95 °C for 10 min. Samples were loaded on 15% SDS-PAGE gels and visualized by Coomassie staining.

*Preparation of branched tri/pentaUbs using sortase.* Generation of branched tri/pentaUbs followed the same protocol as described above for sortase reactions. In short, 5 or 10 µM of Ub-Kx/KyGGK-H$_6$ (in case of preparative assembly Ub-K11/K48GGK-(LPT)-H$_6$) was used as acceptor and 100 µM Ub(LAT) (or 50 µM K48-diUb(LAT*)-H$_6$ (diUb-B) in case of pentaUb assembly) as donor. Srt2A (10 or 20 µM, without H$_6$-tag) was added and the mixture incubated at 37 °C for 1 h. Ni-NTA was used to remove unreacted Ub(LAT) and Srt2A. The Ni-NTA eluate containing the desired branched triUb as well as the two diUb intermediates and the Ub-Kx/KyGGK-(LPT*)-H$_6$ was applied to SEC (50 mM TRIS pH 7.5, 150 mM NaCl) for separation. The fractions containing pure branched triUb (identified by 15% SDS-PAGE) chains were pooled and concentrated using Amicon® centrifugal filter units with a 10 kDa MWCO. Protein concentration was determined as described above. Branched triUb was flash frozen in liquid nitrogen and stored at −80 °C until further use. LC-MS analysis was used to verify the identity of the branched triUb.

Analytical assays for formation of branched triUbs/pentaUbs were performed as described above.

*Preparation of K48-linked tetraUb using sortase.* Assembly of K48-linked tetraUb was performed in an analogous manner as described above for sortase reactions. Taken together, 10 µM of diUb-B was used as donor and 20 µM of diUb-A as acceptor (see Supplementary Fig. 9). Srt2A (5 µM, without H$_6$-tag) was added, followed by incubation at 37 °C for 3 h. Ni-NTA was used to remove Srt2A. The Ni-NTA eluate containing the desired K48-linked tetraUb as well as the two diUbs was applied to SEC (50 mM TRIS pH 7.5, 150 mM NaCl) for separation. The fractions containing pure K48-linked tetraUb (identified by 15% SDS-PAGE) chains were pooled and concentrated using Amicons® (10 kDa MWCO). Protein concentration was determined as described above. K48-linked tetraUb was flash frozen using liquid nitrogen and stored at −80 °C until further use. LC-MS analysis was used to verify the identity of K48-linked tetraUb.

Analytical assays for formation of K48-linked tetraUb were performed as described above.

*Site-specific modification of sfGFP with complex Ub architectures.* Charging of complex architectures on sfGFP was performed with 20 µM of sfGFP-N150GGK-H$_6$ as acceptor and 10 µM branched triUb/K48-linked tetraUb as donor. Srt2A (5 µM, without H$_6$-tag) was added followed by incubation at 37 °C. SDS-PAGE analysis was performed as described above.

### Pull-down assays

*Pull-down assays with GST-Tab2-NZF and GST-hHR23A-UBA2.* Pull-down assays were performed as described previously[14]. In short, 30 µg of GST-tagged Tab2-NZF

or hHR23A-UBA2 were incubated with 30 μL 50% slurry of Glutathione Sepharose 4B (GE Healthcare) preequilibrated with pull-down buffer (50 mM Tris pH 7.5, 150 mM NaCl, 2 mM DTT, 0.1% NP-40) for 1 h at 4 °C. Subsequently the beads were washed four times with pull-down buffer (4000 × g, 2 min, 4 °C) followed by addition of 7 μg of the corresponding K48- or K63-linked diUbs in a total volume of 50 μL (K48-linked diUbs in case of hHR23A-UBA2 and K63-linked diUbs in case of Tab2-NZF). After incubation at 4 °C for 16 h the beads were washed six times with 100 μL pull-down buffer followed by the addition of 50 μL 1× Laemmli buffer and SDS-PAGE analysis.

*Rap80 pull-down assays.* H$_6$-Rap80$_{(1–137)}$-7A-Linker (20 μg) was incubated with 100 μl 50% slurry of Ni-NTA agarose (Jena Bioscience) equilibrated with pull-down buffer (PDB, 50 mM TRIS pH 7.5, 150 mM NaCl, 2 mM TCEP, 0.1% NP-40) at 4 °C for 1 h and subsequently washed three times with PDB (4000 × g, 2 min, 4 °C). Afterwards 50% (v/v) slurry was generated using PDB. To 30 μL of Ni-NTA slurry charged with Rap80$_{(1–137)}$, 3.5 μM of differently linked hybrid chains and respective controls (wt Ub, Srt2A-generated K63-diUb and wt SUMO2) were added in a total volume of 50 μl. After incubation for 1 h at 4 °C, beads were washed five times with 100 μl PDB followed by addition of 50 μl 1× Laemmli buffer. After boiling at 95 °C for 10 min and centrifugation (14,000 × g, 10 min) the samples were loaded on 15% SDS-PAGE gels and visualized by Coomassie staining. Coomassie-stained bands of Rap80 and pulled-down hybrid chains of three independent PD experiments were analyzed densitometrically using the program ImageJ 1.48t (Wayne Rasband, National Institutes of Health, USA, http://imagej.nih.gov/ij). Relative PD efficiencies were calculated by normalizing the densitometric value of pulled-down hybrid chains to the Rap80 densitometric value.

**K$_D$ determination via fluorescence anisotropy.** Experiments were conducted on a Jasco Fluorescence Spectrometer FP-8500 equipped with polarizers (Jasco). Excitation and emission monochromators were set to 490 nm and 520 nm.

Measurements of K48- and K63-linked diUbs were performed in 25 mM sodium phosphate pH 7.4, 150 mM NaCl, 5 mM β-mercaptoethanol, 1 mM EDTA, 0.005% Tween 20. Measurements of hybrid chains were performed in NMR buffer (20 mM Tris pH 6.8, 100 mM NaCl, 4 mM DTT). Total cuvette volume was 60 μL. Concentration of Atto488-labelled Rap80 variants was held constant at 300 nM for all measurements. All data processing was performed using KaleidaGraph software (Synergy Software, Reading, UK). K$_D$ was determined using a single binding site model, and average values and error bars (s.d.) were calculated from three different experiments (n = 3).

**DUB assays.** USP2/UCHL3 was diluted into DUB reaction buffer (25 mM Tris pH 7.5, 150 mM NaCl, 10 mM DTT) and incubated at room temperature for 10 min for activation. Afterwards, 200 nM of USP2/UCHL3 were added to 20 μM Ub variants, displaying different sortase motifs at their C-termini followed by a H$_6$-tag. At denoted time points, 6 μL samples were taken and quenched by the addition of 4× SDS loading buffer. After boiling at 95 °C for 10 min samples were loaded on SDS-PAGE and visualized by Coomassie staining.

**NMR spectroscopy.** NMR data were collected on a Bruker Avance III 900 MHz spectrometer equipped with a cryogenic probe. The chemical shift assignment of backbone resonances for Rap80$_{(35–124)}$ was performed based on heteronuclear experiments such as 1H-15N-HSQC, HNCACB, (H)CC(CO)NH, HNCO, HN(CA)CO[56]. NMR-Spectra were processed using NMRPipe[57] and analyzed using CcpNMR Analysis 2.4.2[58].

NMR experiments were recorded in 20 mM Tris pH 6.8, 100 mM NaCl, 4 mM DTT and 10% D2O at 298 K. Backbone chemical shift assignment, hetNOE, and titration experiment were performed at 180, 90, and 10 μM, respectively. The titration experiments with Rap80$_{(35–124)}$ and the SUMO2-diUb hybrid chains were done as dilution series where the reference and the last titration point (1:1) were measured first and later used to create the 0.5:1 and 0.25:1 titration points.

**Reporting summary.** Further information on research design is available in the Nature Research Reporting Summary linked to this article.

## Data availability

The data generated or analyzed during this study are included in this article (and its supplementary information files) or are available from the corresponding author upon reasonable request. NMR data were deposited at BMRB (https://bmrb.io/) with ID 51092, and are available at https://doi.org/10.13018/BMR51092. Protein structures and models used for the figures are available under the accession codes: 2N9E, 2RR9, 2WWZ, and 1ZO6. Source data are provided with this paper.

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

## Acknowledgements

We thank Marko Cigler for proof reading of the manuscript. This work was supported by the DFG through the following programs: GRK1721 and SFB1309 to K.L. and M.S. and by the European Research Council (ERC) under the European Union's Horizon 2020 research and innovation program (grant agreement no. 101003289—Ubl-tool) to K.L. We thank Anja Bremm, Goethe University Frankfurt for GST-Cdc34, GST-Ubc13, and GST-Uev1A plasmids. We thank Sam Asami for support with NMR experiments and Florent Delhommel for discussions. We thank Matthias Feige for access to Jasco-FP-8500 fluorescence spectrometer.

## Author contributions

M.F. and K.L. conceived the research plan and experimental strategy. M.F. synthesized AzGGK, performed cloning, expression and purification of proteins, the generation and purification of hybrid chains and complex Ub/Ubl architectures as well as pull-down assays and $K_D$-measurements. M.W. created K6-linked diUbs, the K48/K6-linked triUb, performed diUb hydrolysis assays and functional studies on differently linked diUbs. D.S. helped with chemical synthesis, the expression of sortases, E2-enzymes, SUMO variants and the generation of conjugates for hybrid chains as well as for complex Ub architectures. S.G. and M.S. designed and analyzed NMR experiments and data, S.G. performed NMR experiments. All authors analyzed data, M.F. and K.L. wrote the paper with input from the other authors.

## Competing interests

K.L. and M.F. have filed a patent 'Means and methods for site-specific protein modification using transpeptidases', European Patent Application No. 19 745 053.9 – 1118 based on International Application No. PCT/EP2019/067820 that covers generation of Ubl topologies by using orthogonal sortase enzymes. The remaining authors declare no competing interests.
