## [Peer Review File · Nature Communications]

REVIEWER COMMENTS

Reviewer #1 (Remarks to the Author):

In this manuscript, Fottner et. al. report on the use of orthogonal sortase variants to assemble defined ubiquitin chains and generate site-specific polyubiquitylated proteins. The work builds on an exciting paper published in 2019 from the Lang lab in which the unnatural amino acid AzGGK was site-specifically incorporated into a protein of interest to facilitate sortase-mediated site-specific mono-ubiquitylation.

Using hydrolysis assays, the authors identify a pair of sortases, Srt2A and Srt5M, that recognize distinct C-terminal motifs; Srt2A prefers LALTG whereas Srt5M prefers LPLTG. Both variants were previously reported by David Liu's lab. They then go on to show that Srt2A and Srt5M can be used in succession to generate hybrid ubiquitin-SUMO2 chains (Fig. 2 and Supp Fig. 4), homo- and heterotypic ubiquitin chains (Supp. Figs. 3, 6-10), and site-specifically anchor pre-assembled chains to a substrate protein, in this case sfGFP (Fig. 3). Overall, the strategy is quite clever and the data showing the modularity of Srt2A and Srt5M are compelling.

The major concern relates to function. Deep mutational scanning has shown fitness defects arise from substitutions at positions 72 and 74 of ubiquitin (Mavor et. al. 2018 and Mavor et. al. 2016), indicating R72 and R74 are important for downstream activities. Replacing these residues with either alanine and threonine (LALTG) or proline and threonine (LPLTG) could therefore compromise function. The authors show that K63-diUb-SUMO hybrid chains do bind a fragment of RAP80 using qualitative pulldown experiments, suggesting function can be retained despite the C-terminal substitutions. However, there are a couple of concerns with interpretation of these results. First, is that pulldown experiments with ubiquitin-binding domains can lead to artifacts (Sims et. al. Nat. Struct. Mol. Biol. 2009), and thus should be coupled with quantitative binding measurements, e.g., ITC analysis. Second, binding of the sortase-derived chains should be compared to hybrid chains bearing a native C-terminus, which can be generated using K63-specific conjugating enzymes. The functional impact of LALTG and LPLTG C-terminal motifs should also be assessed with some of the other chains that were made, especially considering the purpose is to understand, for example, how the proteasome engages different polyUb-substrates.

Reviewer #2 (Remarks to the Author):

A modular toolbox to generate complex polymeric ubiquitin architectures using orthogonal sortase enzymes

Fottner et al.

Post-translational modification of proteins with ubiquitin (Ub) and Ubls is essential for all eukaryotes and regulates numerous cellular pathways. Recent studies have revealed important roles for complex Ub/Ubl polymers such as SUMO/Ub hybrid chains and heterotypic (or branched) ubiquitin chains. However, detailed characterization of these complex Ub/Ubl chains is limited by the technical difficulty of preparing these chains with defined compositions.

In this manuscript, Lang and coworkers developed a method for producing complex Ub/Ubl polymers using "orthogonal sortylation". First, they established the "Ubl-tools" strategy. In this strategy, they constructed Ub/Ubl chains with a designed topology using two sortase enzymes specific for different recognition motifs and Ub/Ubl mutants containing those sortase recognition motifs. Then, as an application study, they synthesized differently-linked diUb-SUMO2 hybrid chains and investigated their interaction with RAP80. Finally, they showed that the Ubl-tools strategy can be used to build complex Ub chains, such as branched oligomers with defined combinations of bonds.

Overall, the author has established a novel and sophisticated method to build complex Ub/Ubl chains. This has the potential to contribute significantly to the study of Ub/Ubl code biology. However, functional analysis of synthesized Ub/Ubl polymers is less convincing. It is unclear whether the synthesized Ub/Ubl polymers retain the properties of the native polymers (see points 1 and 2). The study will be of great interest provided that the presented method is applicable to functional Ub/Ubl code studies.

Specific concerns:

1. The authors should indicate whether (or to what extent) the synthesized Ub/Ubl chains retain the properties of the native chains. I am concerned because the C-terminal sequence of Ub is involved in various Ub/UBD interactions (e.g., Dikic et al., 2009; PMID 19773779) and is indispensable for Ub function, at least in yeast (Roscoe, et al., 2013; PMID 23376099). The authors' group previously analyzed the interaction of Ub chains derived from the Ub(LAT)/Ub(AT) mutants with several UBDS (Fottner et al., Nat Chem Biol 2019), and the result showed that the K_d value of the interaction between K63-linked diUb(LAT) and RAP80 tUIM is significantly lower than that seen in the native diUb.

2. Along the same line, the characteristics of Ub chains derived from Ub (LPT) have not been analyzed.

For orthogonal sortylation using Srt5M, the C-terminal sequence of Ub has been changed to LPLTGG. However, mutations to proline may affect the structure and/or flexibility of the peptide. The authors should analyze, using a quantitative method, whether the diUb chains derived from the Ub(PT)/Ub(LPT) mutants retain affinity and specificity for decoder proteins (e.g., RAP80, TAB2, RAD23, and linkage-specific DUBs).

3. The authors characterized the interaction of RAP80 with various SUMO2-K63Ub hybrid chains. However, the difference in affinity (Fig. 2f) seems to be marginal. The data may suggest that the SUMO2 conjugation site is not critical for the RAP80 interaction, probably because there is a flexible linker region between tUIM and SIM. The authors should use more quantitative method to analyze the affinity of these hybrid chains for RAP80. Did the authors compare the K_d values of these interactions?

4. Previous studies have shown that heterotypic or branched Ub trimers exhibit unique intramolecular interactions that affect recognition by decoders (Boughton et al., 2019; PMID 31677892). Given the concerns raised in points 1 and 2, it is unclear whether the branched Ub trimers synthesized using the Ubl-tools behave similarly to native Ub trimers.

5. Data presentation. The main figures contain many schematics, and the readers have to look for essential data in the supplementary information. I recommend that the author include some of the important data in the main figure.

6. It seems that some of data is not shown in the manuscript (e.g., P.6, line 145 states 'LC-MS' but I couldn't find the data presentation).

If the authors can address above comments, I am happy to review it again.

Reviewer #3 (Remarks to the Author):

Fottner et al. describe a modular toolbox (Ubl-tools) that allows the stepwise assembly of Ub/Ubl chains in a flexible and user-defined manner facilitated by orthogonal sortase enzymes. The method described does not require advanced chemical expertise and therefore could be implementable in biology research labs. The authors demonstrate the universality and applicability of Ubl-tools by generating complex polymeric Ub/Ubl topologies, including distinctly linked Ub/Ubl hybrid chains, heterotypic and branched chains.

The authors demonstrate the generality of Ubl-tools by building all 7 differently linked diub-SUMO2 hybrid chains and investigation of their binding mode to Rap80. Next they show that Ubl tools can be combined with enzymatic Ub assembly to generate complex Ub-chains and linkage defined branched Ub oligomers.

Overall this is a well written manuscript and the described Ubl-tool technique can provide valuable opportunities for studying the functional impact of these complex type of modifications. This paper will for sure stimulate further investigations into this important aspect and Nature Communications will be the right place to publish this work. With that said, there are a few places in the manuscript that need to be addressed before it is accepted for publication.

- In line 127, the authors describe that prolonged incubation of the diub variants with the sortase enzymes led to diub hydrolysis. According to the authors this is as expected. It would be good to clearly state here that they are referring here to on-target hydrolysis activity displayed at their own recognition motif.
- In line 193 'efficient formation'. Although the methodology reported can be used to generate complex polymeric Ub/Ubl topologies, including distinctly linked Ub/Ubl hybrid chains, heterotypic and branched chains, all gells actually show that the reaction is NOT very efficient. Although a 5-fold excess is used for the acceptor, the reaction needs several days (Suppl figure 4c) and even then reaches a maximum of 30% conversion (judged by eye on their gels). The authors should report yields for their reactions (conversion rates) and discuss also the limitations of the technique.
- As the authors state that the described technology is implementable in biology research labs, they should give more insights into the expression of their starting material (GGk-bearing POIs and Diubs etc) in their methods section. What is typically the scale of their expression? 1L expression, typically yields how many mg's of starting material. As, the authors state that they can obtain mg's of complex Ub-material, it would be good to get a feeling on scale/conversion rates and practicality in general.
- Line 211, please define linear chain (linked via N-terminus of SUMO2).
- To show that the K48-tetra-Ub chains are fully functional (correctly folded), it would be good to perform a DUB cleavage experiment.
- In the discussion section the authors discuss a future application (line 337-346) in proteomic identification of interacting proteins or receptor proteins. However, a limitation here would be hydrolysis of the Ub/Ubl chain. The linkages described have isopeptide bonds which are prone to

hydrolysis. How would the authors circumvent that? The chains reported here would not be suited for such an application. It would be good if the authors could discuss this in more detail.

- With the orthogonal sortase enzymes described here the authors can generate complex Ub/Ubl architectures. It would be nice to also comment on Ubl/Ubl architectures in the discussion section. Ubiquitinated SUMO1-3 chains have also been reported. The C-terminus of SUMO is different than for Ub, Nedd8 and ISG15 (QQQTGG vs LXLGG). Can these sortases also be implemented on SUMO to form SUMO-SUMO-Ub chains?

REVIEWER COMMENTS

Reviewer #1 (Remarks to the Author):

In this manuscript, Fottner et. al. report on the use of orthogonal sortase variants to assemble defined ubiquitin chains and generate site-specific polyubiquitylated proteins. The work builds on an exciting paper published in 2019 from the Lang lab in which the unnatural amino acid AzGGK was site-specifically incorporated into a protein of interest to facilitate sortase-mediated site-specific mono-ubiquitylation.

Using hydrolysis assays, the authors identify a pair of sortases, Srt2A and Srt5M, that recognize distinct C-terminal motifs; Srt2A prefers LALTG whereas Srt5M prefers LPLTG. Both variants were previously reported by David Liu's lab. They then go on to show that Srt2A and Srt5M can be used in succession to generate hybrid ubiquitin-SUMO2 chains (Fig. 2 and Supp Fig. 4), homo- and heterotypic ubiquitin chains (Supp. Figs. 3, 6-10), and site-specifically anchor pre-assembled chains to a substrate protein, in this case sfGFP (Fig. 3). Overall, the strategy is quite clever and the data showing the modularity of Srt2A and Srt5M are compelling.

We thank the reviewer for their enthusiasm for our work.

The major concern relates to function. Deep mutational scanning has shown fitness defects arise from substitutions at positions 72 and 74 of ubiquitin (Mavor et. al. 2018 and Mavor et. al. 2016), indicating R72 and R74 are important for downstream activities. Replacing these residues with either alanine and threonine (LALTG) or proline and threonine (LPLTG) could therefore compromise function.

We thank the reviewer for their comment. To study and experimentally validate structural and functional integrity of Ub variants bearing the different sortase motifs (for Srt2A: AT and LAT; for Srt5M: PT and LPT), we have built K63- and K48-linked diUbs bearing these recognition motifs in the linker region connecting the two Ub monomers (K63- and K48-linked diUb(AT), diUb(LAT), diUb(PT) and diUb(LPT)) and tested whether they are selectively recognized by specific Ub-binding domains (UBDs). We tested them against a K63-linkage specific antibody, performed in vitro pulldown assays with a UBD specific for K63 chains (protein kinase TAK1 adaptor subunit TAB2-NZF) as well as a UBD specific for K48 chains (proteasomal shuttling factor hHR23A-UBA2). Furthermore, we measured binding affinities (KDs) for all K63-linked diUbs towards the K63 linkage-sensitive Rap80-tandem Ub-interacting motifs (tUIMs).

The K63-specific antibody recognized all five K63-diUbs (wt K63-diUb, K63-diUb(AT), K63-diUb(LAT), K63-diUb(PT) and K63-diUb(LPT)) at similar levels, indicating that AT, LAT, PT and LPT substitutions in the linker region do not interfere with binding to the linkage-specific antibody. For binding to the different UBDs (TAB2-NZF, hHR23A-UBA2 and Rap80-tUIMs) we observed wt-like behavior for AT, LAT and LPT mutations, while the PT linker showed more compromised binding, indicating indeed that the R74P substitution might be less optimal for mimicking wt Ub chain behavior and that it may be beneficial to introduce the leucine spacer amino acid. In conclusion, we could however show that both Srt2A and Srt5M-generated diUbs largely retain their binding affinity towards linkages specific UBDs, a requirement for triggering diverse cellular signaling events, including protein kinase activation, DNA-damage repair and protein degradation.

We have inserted a new Fig 2 (accompanied by Supplementary Figs 3 and 4) into the manuscript and have added the following paragraph describing the results of these experiments (highlighted in blue in the manuscript).

A crucial determinant for Ub-mediated cellular signaling consists in the ability of distinct effector proteins to convert distinct Ub patterns into specific functional outcomes.(ref) To experimentally validate the functional and structural integrity of our sortase-generated linkages, we set out to build K63- and K48-linked diUbs bearing the different Srt2A and Srt5M motifs in the linker region connecting the two Ub monomers (diUb(AT), diUb(LAT), diUb(PT) and diUb(LPT)) and tested whether they are selectively recognized by specific Ub-binding domains (UBDs). First, we incubated these sortase-generated variants as well as their respective wt counterparts (i.e. wt K63-diUb and wt K48-diUb) with a K63-linkage specific antibody (Fig. 2a and Supplementary Fig. 4a). All K63-linked diUbs were recognized to a similar extent to natively linked K63-diUb in anti-K63 western blots, indicating that AT, LAT, PT or LPT substitutions in the linker region do not interfere with binding to the linkage-specific antibody. As expected, all K48-linked diUbs did not bind to the K63-linkage specific antibody. Next, we probed K63-linked diUbs in in vitro pulldown (PD) assays with protein kinase TAK1 adaptor subunit TAB2, which contains an Npl4 zinc finger (NZF) UBD that specifically senses K63-linked chains (Fig. 2b and Supplementary Fig 4b).(ref) K63-diUb(AT), K63-diUb(LAT) and K63-diUb(LPT) retained the ability to bind TAB2-NZF, while K63-linked diUb(PT) containing the PT linker between the two Ub monomers showed compromised binding, indicating that the R74P substitution might be less optimal for mimicking wt K63-diUb behavior. Similarly, also the more compact sortase-generated K48-diUbs displaying AT, LAT and LPT substitutions retained their ability to bind the designated Ub-associated (UBA) domain of proteasomal shuttling factor hHR23A-UBA2 in in vitro PD assays (Fig. 2c and Supplementary Fig. 4c), while K48-diUb(PT) failed to properly bind to hHR23A-UBA2. To study binding properties of different sortase-generated diUbs in more quantitative terms, we determined binding constants of differently linked K63-diUbs and a fluorescently labeled Rap80 construct that harbors K63-sensitive tandem Ub-interacting motifs (tUIMs) via fluorescence anisotropy (Rap80-tUIMs(79-124), Fig 2d).(ref) For all four sortase-generated K63-diUbs we measured distinct binding affinities. KDs for AT-, LAT- and LPT- linked diUbs were slightly lower as for wt-K63-diUb (approximately two-fold reduction for diUb(AT) and four- to five-fold reduction for diUb(LAT) and diUb(LPT), respectively). K63-diUb(PT) displayed a 10-fold lower binding affinity towards Rap80-tUIMs (Fig. 2d), confirming our previous observation that the proline mutation at position 72 might give diUbs an unusual conformational rigidity. This indicates that the PT linker may be less optimal than the other investigated sortase linkers for recognition by some UBDs and that it might therefore be beneficial to introduce the leucine spacer amino acid to resemble more wt-like behavior. Nevertheless, we could show that both Srt2A- and Srt5M-generated diUbs largely retain their binding affinity towards linkage specific UBDs, a requirement for triggering diverse cellular signaling events.

The authors show that K63-diUb-SUMO hybrid chains do bind a fragment of RAP80 using qualitative pulldown experiments, suggesting function can be retained despite the C-terminal substitutions. However, there are a couple of concerns with interpretation of these results. First, is that pulldown experiments with ubiquitin-binding domains can lead to artifacts (Sims et. al. Nat. Struct. Mol. Biol. 2009), and thus should be coupled with quantitative binding measurements, e.g., ITC analysis. Second, binding of the sortase-derived chains should be

compared to hybrid chains bearing a native C-terminus, which can be generated using K63-specific conjugating enzymes. The functional impact of LALTG and LPLTG C-terminal motifs should also be assessed with some of the other chains that were made, especially considering the purpose is to understand, for example, how the proteasome engages different polyUb-substrates.

We thank the reviewer for their comment. We have complemented our pulldown experiments with further biophysical measurements to investigate binding affinities and structural features of the binding modes of differently linked diUb-SUMO2 hybrid chains towards Rap80. We have performed both fluorescence anisotropy using fluorescently labeled Rap80 as well as NMR titration experiments using ¹⁵N-labeled Rap80 with three differently linked hybrid chains.

As suggested by the reviewer, we have built hybrid chains by combining enzymatic assembly of endogenous K63-diUb with sortylation using Srt5M and the 'LPT' sortase recognition motif. These experiments corroborate the novel binding mode involving the K21-linked hybrid chain and Rap80 that relies on contributions of the ~30 amino acid long linker to enable simultaneous binding of SIM and tUIMs. We propose that the K21-linked diUb-SUMO2 chain forces this highly positively charged linker into a kinked conformation and localizes it near a negatively charged region at the SUMO2 surface adjacent to the hydrophobic SIM-binding groove.

We have split original Fig. 2 into new Fig 3 and Fig 4 that now contain data regarding the assembly and the characterization of diUb-SUMO hybrid chains and their binding to Rap80. Additional data can be found in Supplementary Figs 6-8.

We have added the following paragraph to the manuscript (highlighted in blue in the manuscript):

In order to investigate binding affinities and structural features of the binding modes of different hybrid chains towards Rap80, we set out to determine binding constants via fluorescence anisotropy using fluorescently labeled Rap80 and to perform NMR titration experiments using ¹⁵N-labeled Rap80 with three differently linked hybrid chains. To best resemble the functional and structural integrity of endogenous K63-diUb-SUMO2 hybrid chains, we built hybrid chains by combining enzymatic assembly of K63-linked diUb with sortylation using Srt5M (Fig. 3f). For this, we first assembled natively linked K63-diUb bearing a Srt5M motif (LPT) at its proximal Ub-C-terminus using the E1 enzyme UBE1 and the K63-linkage specific E2 enzymes Ubc13 and Uev1a (Supplementary Fig. 7b). In order to guarantee distinct formation of this diUb variant, we incubated unmodified Ub with an Ub-variant bearing a Srt5M recognition motif (LLPLTG) lacking the C-terminal glycine G76, followed by a short linker sequence (LHG YEAAAK). Omitting G76 that is not needed for Srt5M recognition guarantees orthogonality towards E1 and E2. The short linker assures good sortase accessibility and contains a masked Ni²⁺-binding peptide (GLHG) that boosts sortylation efficiency by inactivating the emerging nucleophile through Ni²⁺ complex formation (Supplementary Fig. 7c)43. Using this approach, we built linear, K21- and K42-linked diUb-SUMO2 hybrid chains containing the wt linker between the two K63-linked Ubs and the LPT linker between K63-diUb and SUMO2 in good yields using K63-diUb and SUMO2-GGK in equimolar ratios (Fig. 3f and Supplementary Fig 7c,d).

We first determined binding constants of these three hybrid chains using a fluorescently labeled Rap80 construct (Rap80(35-124)) using fluorescence anisotropy (Fig. 4c, Supplementary Fig. 8a). In excellent agreement with our PD-data we observed three times tighter Rap80-binding for the K21-linked hybrid chain (K_D = 0.4 μM) than for the linearly-linked

diUb-SUMO2 ($KD = 1.2 \mu M$). Correspondingly, the K42-linked chain showed 6-fold weaker binding ($KD = 2.5 \mu M$) than the K21-linked chain. To characterize the structural features of the different binding modes of these *diUb-SUMO2* chains towards Rap80, we expressed and purified uniformly ^{15}N -labeled Rap80(35-124) (Supplementary Fig. 8a). The secondary structure of Rap80 was analyzed based on $^{13}C\alpha$ and $^{13}C\beta$ secondary chemical shifts and reveals that Rap80 shows an unstructured region from amino acid 34 to 56, shows a helical propensity for residues 62 to 79 and an α -helical region from amino acid 79 to 122 (Supplementary Fig 8c). Titrating the three differently linked unlabeled *diUb-SUMO2* chains to this construct, led to differential line broadening, shown by signal intensity reductions $I/I(\text{ref})$, of the backbone amide resonances of the tUIMs and SIM in Rap80, indicating binding of *diUb* and SUMO2 to these regions, as expected (Fig 4d). Notably, we also observed line-broadening in the unstructured linker region (amino acids 47-62) connecting SIM and tUIMs, suggesting an additional binding interface involving this linker segment (Fig. 4d). Strikingly, this effect was most pronounced for the K21-linked hybrid chain, as observed by stronger signal intensity reductions in the linker region, especially for the positively charged region around lysine K61 (residues 59-75, Fig. 4e). Moreover, the SIM binding interface is extended and involves also residues around the archetypical hydrophobic β -strand (F40IVI). In contrast, titrations with the K42-linked chain, led to a drop in signal intensity only within the F40IVI core motif, while flanking regions and residues in the subsequent linker (residues 47-62) were largely unaffected. Furthermore, with the K42-linked chain, we observed appearance of a second low populated set of peaks for residues F40IVI within the SIM- β -strand (Fig. 4e). This is consistent with either parallel or antiparallel β -strand binding to SUMO as suggested previously (Supplementary Fig. 8d)⁴² and corroborates that the K42-linked hybrid chain hinders an optimal binding of the SIM and linker region, reflected by its overall lower binding affinity.

Thus, the NMR data and binding affinity measurements suggest a novel binding mode between *diUb-SUMO2* hybrid chains and Rap80 that relies on contributions of the ~30 amino acid long linker to enable simultaneous accessibility of SIM and tUIMs. We propose that the K21-linked *diUb-SUMO2* hybrid chain forces this linker into a kinked conformation and localizes it near a negatively charged region at the SUMO2 surface adjacent to the hydrophobic SIM-binding groove (Fig. 4f-g). Charge complementarity involving Rap80 linker residues 59-75 and the negatively charged groove on SUMO2 may thus further enhance the Rap80/SUMO2 interaction, consistent with the higher binding affinity compared to the linear and K42-linked hybrid chain (Fig. 4 and Supplementary Fig. 8).

Reviewer #2 (Remarks to the Author):

A modular toolbox to generate complex polymeric ubiquitin architectures using orthogonal sortase enzymes

Fottner et al.

Post-translational modification of proteins with ubiquitin (Ub) and Ubls is essential for all eukaryotes and regulates numerous cellular pathways. Recent studies have revealed important roles for complex Ub/Ubl polymers such as SUMO/Ub hybrid chains and heterotypic (or branched) ubiquitin chains. However, detailed characterization of these complex Ub/Ubl chains is limited by the technical difficulty of preparing these chains with defined compositions. In this manuscript, Lang and coworkers developed a method for producing complex Ub/Ubl

polymers using "orthogonal sortylation". First, they established the "Ubl-tools" strategy. In this strategy, they constructed Ub/Ubl chains with a designed topology using two sortase enzymes specific for different recognition motifs and Ub/Ubl mutants containing those sortase recognition motifs. Then, as an application study, they synthesized differently-linked diUb-SUMO2 hybrid chains and investigated their interaction with RAP80. Finally, they showed that the Ubl-tools strategy can be used to build complex Ub chains, such as branched oligomers with defined combinations of bonds.

Overall, the author has established a novel and sophisticated method to build complex Ub/Ubl chains. This has the potential to contribute significantly to the study of Ub/Ubl code biology. However, functional analysis of synthesized Ub/Ubl polymers is less convincing. It is unclear whether the synthesized Ub/Ubl polymers retain the properties of the native polymers (see points 1 and 2). The study will be of great interest provided that the presented method is applicable to functional Ub/Ubl code studies.

We thank the reviewer for their overall positive evaluation of our approach and manuscript.

Specific concerns:

1. The authors should indicate whether (or to what extent) the synthesized Ub/Ubl chains retain the properties of the native chains. I am concerned because the C-terminal sequence of Ub is involved in various Ub/UBD interactions (e.g., Dikic et al., 2009; PMID 19773779) and is indispensable for Ub function, at least in yeast (Roscoe, et al., 2013; PMID 23376099). The authors' group previously analyzed the interaction of Ub chains derived from the Ub(LAT)/Ub(AT) mutants with several UBDs (Fottner et al., Nat Chem Biol 2019), and the result showed that the Kd value of the interaction between K63-linked diUb(LAT) and RAP80 tUIM is significantly lower than that seen in the native diUb.

2. Along the same line, the characteristics of Ub chains derived from Ub (LPT) have not been analyzed. For orthogonal sortylation using Srt5M, the C-terminal sequence of Ub has been changed to LPLTGG. However, mutations to proline may affect the structure and/or flexibility of the peptide. The authors should analyze, using a quantitative method, whether the diUb chains derived from the Ub(PT)/Ub(LPT) mutants retain affinity and specificity for decoder proteins (e.g., RAP80, TAB2, RAD23, and linkage-specific DUBs).

We thank the reviewer for these comments (in agreement with comment 1 of reviewer 1).

To study and experimentally validate structural and functional integrity of Ub topologies bearing sortase recognition motifs in the linker region, we have built all possible K48- and K63-linked diUbs (wt-diUb, diUb(AT), diUb(LAT), diUb(PT) and diUb(LPT)) and tested whether they are still recognized by specific Ub-binding domains (UBDs). We tested them against a K63-linkage specific antibody, performed in vitro pulldown assays with a UBD specific for K63 chains (protein kinase TAK1 adaptor subunit TAB2-NZF) as well as a UBD specific for K48 chains (proteasomal shuttling factor hHR23A-UBA2). Furthermore, we measured binding affinities (KDs) for all K63-linked diUbs towards the K63 linkage-sensitive Rap80-tandem Ub-interacting motifs (tUIMs). The K63-specific antibody recognized all five K63-diUbs (wt K63-diUb, K63-diUb(AT), K63-diUb(LAT), K63-diUb(PT) and K63-diUb(LPT)) at similar levels, indicating that AT, LAT, PT and LPT substitutions in the linker region do not interfere with binding to the linkage-specific antibody. For binding to the different UBDs (TAB2-NZF, hHR23A-UBA2 and Rap80-tUIMs) we observed close to wt-like behavior for AT, LAT and LPT mutations, while the PT linker showed more compromised binding, indicating indeed that the R74P substitution might be less optimal for mimicking wt Ub chain behavior and that it may be beneficial to introduce the leucine spacer amino acid. In conclusion, we could however

show that both Srt2A and Srt5M-generated diUbs largely retain their binding affinity towards linkages specific UBDs, a requirement for triggering diverse cellular signaling events, including protein kinase activation, DNA-damage repair and protein degradation.

We have split original Fig. 2 into new Fig 3 and Fig 4 that now contain data regarding the building and the characterization of diUb-SUMO hybrid chains and their binding to Rap80. Additional data can be found in Supplementary Figs 6-8.

We have added the following paragraph to the manuscript (highlighted in blue in the manuscript):

To experimentally validate the functional and structural integrity of our sortase-generated linkages, we set out to build K63- and K48-linked diUbs bearing the different Srt2A and Srt5M motifs in the linker region connecting the two Ub monomers (diUb(AT), diUb(LAT), diUb(PT) and diUb(LPT)) and tested whether they are selectively recognized by specific Ub-binding domains (UBDs). First, we incubated these sortase-generated variants as well as their respective wt counterparts (i.e. wt K63-diUb and wt K48-diUb) with a K63-linkage specific antibody (Fig. 2a and Supplementary Fig. 4a). All K63-linked diUbs were recognized to a similar extent to natively linked K63-diUb in anti-K63 western blots, indicating that AT, LAT, PT or LPT substitutions in the linker region do not interfere with binding to the linkage-specific antibody. As expected, all K48-linked diUbs did not bind to the K63-linkage specific antibody. Next, we probed K63-linked diUbs in in vitro pulldown (PD) assays with protein kinase TAK1 adaptor subunit TAB2, which contains an Npl4 zinc finger (NZF) UBD that specifically senses K63-linked chains (Fig. 2b and Supplementary Fig 4b). (ref) K63-diUb(AT), K63-diUb(LAT) and K63-diUb(LPT) retained the ability to bind TAB2-NZF, while K63-linked diUb(PT) containing the PT linker between the two Ub monomers showed compromised binding, indicating that the R74P substitution might be less optimal for mimicking wt K63-diUb behavior. Similarly, also the more compact sortase-generated K48-diUbs displaying AT, LAT and LPT substitutions retained their ability to bind the designated Ub-associated (UBA) domain of proteasomal shuttling factor hHR23A-UBA2 in in vitro PD assays (Fig. 2c and Supplementary Fig. 4c), while K48-diUb(PT) failed to properly bind to hHR23A-UBA2. To study binding properties of different sortase-generated diUbs in more quantitative terms, we determined binding constants of differently linked K63-diUbs and a fluorescently labeled Rap80 construct that harbors K63-sensitive tandem Ub-interacting motifs (tUIMs) via fluorescence anisotropy (Rap80-tUIMs(79-124), Fig 2d). (ref) For all four sortase-generated K63-diUbs we measured distinct binding affinities. KDs for AT-, LAT- and LPT- linked diUbs were slightly lower as for wt-K63-diUb (approximately two-fold reduction for diUb(AT) and four- to five-fold reduction for diUb(LAT) and diUb(LPT), respectively). K63-diUb(PT) displayed a 10-fold lower binding affinity towards Rap80-tUIMs (Fig. 2d), confirming our previous observation that the proline mutation at position 72 might give diUbs an unusual conformational rigidity. This indicates that the PT linker may be less optimal than the other investigated sortase linkers for recognition by some UBDs and that it might therefore be beneficial to introduce the leucine spacer amino acid to resemble more wt-like behavior. Nevertheless, we could show that both Srt2A- and Srt5M-generated diUbs largely retain their binding affinity towards linkage specific UBDs, a requirement for triggering diverse cellular signaling events.

3. The authors characterized the interaction of RAP80 with various SUMO2-K63Ub hybrid chains. However, the difference in affinity (Fig. 2f) seems to be marginal. The data may suggest that the SUMO2 conjugation site is not critical for the RAP80 interaction, probably

because there is a flexible linker region between tUIM and SIM. The authors should use more quantitative method to analyze the affinity of these hybrid chains for RAP80. Did the authors compare the Kd values of these interactions?

We thank the reviewer for their comment, which coincides with comment 2 of reviewer 1. We have complemented our pulldown experiments with further biophysical measurements to investigate binding affinities and structural features of the binding modes of differently linked diUb-SUMO2 hybrid chains towards Rap80. We have determined KDs via fluorescence anisotropy using fluorescently labeled Rap80 and have performed NMR titration experiments using ¹⁵N-labeled Rap80 with three differently linked hybrid chains.

In addition, to best resemble the functional and structural integrity of endogenous K63-diUb-SUMO2 hybrid chains, we build hybrid chains by combining enzymatic assembly of K63-linked diUb with sortylation using Srt5M (using the 'LPT' sortase recognition motif). In excellent agreement with our pulldown experiments using the sortase-generated hybrid chains with 'AT' and 'PT' mutations, we observed tighter Rap80 binding for the K21-linked hybrid chain as compared to the linearly linked chain. Correspondingly, the K42-linked chains showed 6-fold weaker binding than the K21-linked chain. To characterize the structural features of the different binding modes of these diUb-SUMO2 chains towards Rap80, we teamed up with the lab of Michael Sattler and performed NMR titration experiments using ¹⁵N-labeled Rap80. These experiments corroborate the novel binding mode involving the K21-linked hybrid chain and Rap80 that relies on contributions of the ~30 amino acid long linker to enable simultaneous binding of SIM and tUIMs. We propose that the K21-linked diUb-SUMO2 chain forces this highly positively charged linker into a kinked conformation and localizes it near a negatively charged region at the SUMO2 surface adjacent to the hydrophobic SIM-binding groove.

We have split original Fig. 2 into new Fig 3 and Fig 4 that now contain data regarding the building and the characterization of diUb-SUMO hybrid chains and their binding to Rap80. Additional data can be found in Supplementary Figs 6-8. We have added the following paragraph to the manuscript (highlighted in blue in the manuscript):

We first determined binding constants of these three hybrid chains using a fluorescently labeled Rap80 construct (Rap80(35-124)) using fluorescence anisotropy (Fig. 4c, Supplementary Fig. 8a). In excellent agreement with our PD-data we observed three times tighter Rap80-binding for the K21-linked hybrid chain (KD = 0.4 μM) than for the linearly-linked diUb-SUMO2 (KD = 1.2 μM). Correspondingly, the K42-linked chain showed 6-fold weaker binding (KD = 2.5 μM) than the K21-linked chain. To characterize the structural features of the different binding modes of these diUb-SUMO2 chains towards Rap80, we expressed and purified uniformly ¹⁵N-labeled Rap80(35-124) (Supplementary Fig. 8a). The secondary structure of Rap80 was analyzed based on ¹³Cα and ¹³Cβ secondary chemical shifts and reveals that Rap80 shows an unstructured region from amino acid 34 to 56, shows a helical propensity for residues 62 to 79 and an α-helical region from amino acid 79 to 122 (Supplementary Fig 8c). Titrating the three differently linked unlabeled diUb-SUMO2 chains to this construct, led to differential line broadening, shown by signal intensity reductions I/I(ref), of the backbone amide resonances of the tUIMs and SIM in Rap80, indicating binding of diUb and SUMO2 to these regions, as expected (Fig 4d). Notably, we also observed line-

broadening in the unstructured linker region (amino acids 47-62) connecting SIM and tUIMs, suggesting an additional binding interface involving this linker segment (Fig. 4d). Strikingly, this effect was most pronounced for the K21-linked hybrid chain, as observed by stronger signal intensity reductions in the linker region, especially for the positively charged region around lysine K61 (residues 59-75, Fig. 4e). Moreover, the SIM binding interface is extended and involves also residues around the archetypical hydrophobic β -strand (F40IVI). In contrast, titrations with the K42-linked chain, led to a drop in signal intensity only within the F40IVI core motif, while flanking regions and residues in the subsequent linker (residues 47-62) were largely unaffected. Furthermore, with the K42-linked chain, we observed appearance of a second low populated set of peaks for residues F40IVI within the SIM- β -strand (Fig. 4e). This is consistent with either parallel or antiparallel β -strand binding to SUMO as suggested previously (Supplementary Fig. 8d)42 and corroborates that the K42-linked hybrid chain hinders an optimal binding of the SIM and linker region, reflected by its overall lower binding affinity.

Thus, the NMR data and binding affinity measurements suggest a novel binding mode between diUb-SUMO2 hybrid chains and Rap80 that relies on contributions of the ~30 amino acid long linker to enable simultaneous accessibility of SIM and tUIMs. We propose that the K21-linked diUb-SUMO2 hybrid chain forces this linker into a kinked conformation and localizes it near a negatively charged region at the SUMO2 surface adjacent to the hydrophobic SIM-binding groove (Fig. 4f-g). Charge complementarity involving Rap80 linker residues 59-75 and the negatively charged groove on SUMO2 may thus further enhance the Rap80/SUMO2 interaction, consistent with the higher binding affinity compared to the linear and K42-linked hybrid chain (Fig. 4 and Supplementary Fig. 8).

4. Previous studies have shown that heterotypic or branched Ub trimers exhibit unique intramolecular interactions that affect recognition by decoders (Boughton et al., 2019; PMID 31677892). Given the concerns raised in points 1 and 2, it is unclear whether the branched Ub trimers synthesized using the Ubi-tools behave similarly to native Ub trimers.

We think we were able to address the concerns raised in point 1 and 2 by showing that diUbs bearing AT, LAT and LPT mutations behave largely as their wt-counterparts in binding the chosen UBDs. For PT mutations we have seen more compromised binding to specific UBDs, indicating that substitution of R72 (the first amino acid succeeding the C-terminal β -sheet in Ub) to proline might give an unusual conformational rigidity to Ub chains compromising their binding to some UBDs. It might therefore be beneficial to use the Srt5M 'LPT' recognition motif including the leucine spacer amino acid.

To unambiguously test if branched Ub trimers behave as native Ub trimers it would have been necessary to perform NMR or SANS experiments as in the mentioned publication Boughton et al., 2019; PMID 31677892 and compare the data to natively linked branched triUb.

Given the focus of our manuscript, we instead decided to focus on Rap80-binding diUb-SUMO2 hybrid chains and performed fluorescence anisotropy as well as NMR-titrations on these hybrid chains. We nevertheless think that analysis of triUbs will be an interesting future endeavor.

5. Data presentation. The main figures contain many schematics, and the readers have to look for essential data in the supplementary information. I recommend that the author include some of the important data in the main figure.

We agree with this and thank the reviewer for this comment. We have now rearranged some of the data, so that the manuscript has in total 5 figures that show next to the schemes, which we think are important for understanding the concept of Ubl-tools, more primary data. These 5 figures are accompanied by 13 detailed supplementary figures.

6. It seems that some of data is not shown in the manuscript (e.g., P.6, line 145 states 'LC-MS' but I couldn't find the data presentation).

We have double checked that all mentioned data is shown in figures. The LC-MS under question was part of Supplementary Fig. 3 in the original submission. In rearranging figures we have moved this LC-MS to main Fig. 3a.

If the authors can address above comments, I am happy to review it again.

Reviewer #3 (Remarks to the Author):

Fottner et al. describe a modular toolbox (Ubl-tools) that allows the stepwise assembly of Ub/Ubl chains in a flexible and user-defined manner facilitated by orthogonal sortase enzymes. The method described does not require advanced chemical expertise and therefore could be implementable in biology research labs. The authors demonstrate the universality and applicability of Ubl-tools by generating complex polymeric Ub/Ubl topologies, including distinctly linked Ub/Ubl hybrid chains, heterotypic and branched chains. The authors demonstrate the generality of Ubl-tools by building all 7 differently linked diub-SUMO2 hybrid chains and investigation of their binding mode to Rap80. Next they show that Ubl tools can be combined with enzymatic Ub assembly to generate complex Ub-chains and linkage defined branched Ub oligomers.

Overall this is a well written manuscript and the described Ubl-tool technique can provide valuable opportunities for studying the functional impact of these complex type of modifications. This paper will for sure stimulate further investigations into this important aspect and Nature Communications will be the right place to publish this work.

We thank the reviewer for their enthusiasm for our approach and positive evaluation of our work.

With that said, there are a few places in the manuscript that need to be addressed before it is accepted for publication.

- In line 127, the authors describe that prolonged incubation of the diub variants with the sortase enzymes led to diub hydrolysis. According to the authors this is as expected. It would be good to clearly state here that they are referring here to on-target hydrolysis activity displayed at their own recognition motif.

We adjusted this sentence and made it clearer. It reads now as following:

As expected, prolonged incubation of a diUb variant with its own sortase enzyme, i.e. the sortase variant that was used to assemble it and therefore displays the target recognition motif, led to on-target diUb hydrolysis yielding the two corresponding Ub-monomers.

- In line 193 'efficient formation'. Although the methodology reported can be used to generate complex polymeric Ub/Ubl topologies, including distinctly linked Ub/Ubl hybrid chains,

heterotypic and branched chains, all gells actually show that the reaction is NOT very efficient. Although a 5-fold excess is used for the acceptor, the reaction needs several days (Suppl figure 4c) and even then reaches a maximum of 30% conversion (judged by eye on their gels). The authors should report yields for their reactions (conversion rates) and discuss also the limitations of the technique.

We have now determined all sortylation yields densitometrically and noted them in the corresponding figures. As expected, the yields are dependent on accessibility of the positions that have to be ubiquitylated. Furthermore, somewhat counter-intuitively for sortase-transpeptidations, in our approach the GGK-bearing protein (mimicking the attacking GG-nucleophile) is the 'more precious' component and is therefore used in limiting amounts (as opposed to small molecule/peptide sortase labeling reactions where the GG-bearing nucleophile is used in very high excess).

Nevertheless, yields for diverse Ub chain formations (including all built branched and heterotypically linked chains, as well as charging Ub chains onto POIs) reach yields between 19 % (in the worst case) and up to 77 %. This compares very favorably to enzymatic assembly (please note that many of the Ub chains generated with Ubl-tools are not accessible via existing methods).

Furthermore, we have now shown that sortylation yields can be boosted (even by using equimolar stoichiometries of GGK-POI and sortase motif-bearing Ubl) by applying Ni²⁺ mediated selective nucleophile quenching (see Supplementary Fig. 7c)

- As the authors state that the described technology is implementable in biology research labs, they should give more insights into the expression of their starting material (GGk-bearing POIs and Diubs etc) in their methods section. What is typically the scale of their expression? 1L expression, typically yields how many mg's of starting material. As, the authors state that they can obtain mg's of complex Ub-material, it would be good to get a feeling on scale/conversion rates and practicality in general.

Thanks for this comment. We have added all this information to the online and supplementary methods.

Typical expression scale with AzGGK was 1 L. Depending on the position of the TAG codon approx. 5 – 20 mg/L SUMO, 5 – 15 mg/L Ub and 20 – 40 mg/L GFP were isolated.

- Line 211, please define linear chain (linked via N-terminus of SUMO2).

Thanks for the comment. We have fixed it.

- To show that the K48-tetra-Ub chains are fully functional (correctly folded), it would be good to perform a DUB cleavage experiment.

As noted in the revised manuscript, sortase-generated Ub chains (via AT, LAT, PT and LPT mutations) are refractory to DUB cleavage (shown for USP2 and UCHL3). We have included a respective paragraph and Supplementary Fig. 3. Resistance to DUB hydrolysis constitutes an important feature for Ubl-tools as it allows to use the generated Ub/Ubl topologies for identifying Ub chain-specific interactor proteins in cell lysates and provides valuable tools for interrogating cell-signaling pathways.

Furthermore, we have probed the K48-tetraUb with USP2 and have observed the expected hydrolysis pattern (see Supplementary Fig. 10c). An advantage of Ubl-tools consists in the fact that we can strategically combine orthogonal sortylation with enzymatic diUb formation,

which enables the generation of complex topologies where we can place DUB-resistant and DUB-susceptible linkages at defined positions.

We have added the following paragraphs to the manuscript (highlighted in blue). The corresponding data can be found in Supplementary Fig.3 and Supplementary Fig. 10c.

Having established an orthogonal sortase pair for Ubl-tools, we first examined if Ub variants with correspondingly modified C-termini were still substrates for deubiquitylases (DUBs). While we observed complete cleavage of a C-terminal hexa-histidine H₆-tag upon incubation of wt-Ub-H₆ with the catalytic domain of USP2 or with UCHI3, Ub variants bearing a C-terminal H₆-tag succeeding the Srt2A ('AT' or 'LAT') or Srt5M ('PT' or 'LPT') motif were refractory to DUB cleavage (Supplementary Fig. 3). Resistance to DUB hydrolysis constitutes an important feature for Ubl-tools as it allows to use the generated Ub/Ubl topologies for identifying Ub chain-specific interactor proteins in cell lysates and provides valuable tools for interrogating cell-signaling pathways.

The K48-linked tetraUb contains two wt linkages, as well as a Srt2A-motif linkage (LAT) and bears the Srt5M-motif (LPT) at its C-terminus. Incubation with the hydrolase USP2 resulted in the expected pattern, with both LAT and LPT linkages being refractory to DUB cleavage, while wt linkages are completely hydrolyzed by USP2 (Supplementary Fig. 9c).

- In the discussion section the authors discuss a future application (line 337-346) in proteomic identification of interacting proteins or receptor proteins. However, a limitation here would be hydrolysis of the Ub/Ubl chain. The linkages described have isopeptide bonds which are prone to hydrolysis. How would the authors circumvent that? The chains reported here would not be suited for such an application. It would be good if the authors could discuss this in more detail. See comment above. We have now added the DUB hydrolysis data that show that our sortase-generated linkages (with Srt2A and Srt5M) are recalcitrant to DUB cleavage, making them indeed valuable tools for identifying Ub chain-specific interactor proteins in cell lysates and provides valuable tools for interrogating cell-signaling pathways.

- With the orthogonal sortase enzymes described here the authors can generate complex Ub/Ubl architectures. It would be nice to also comment on Ubl/Ub architectures in the discussion section. Ubiquitinated SUMO1-3 chains have also been reported. The C-terminus of SUMO is different then for Ub, Nedd8 and ISG15 (QQQTGG vs LXLRRGG). Can these sortases also be implemented on SUMO to form SUMO-SUMO-Ub chains?

We have shown previously (Fottner et al NatChemBio 2019) that sortylation is also compatible with SUMOylation and we envision that it can therefore also be used to build SUMO-SUMO-Ub chains. Also other Ubls such as Nedd8 and ISG15 will only require mutations of 1 or 2 residues in their C-terminus to be accepted by Srt2A or Srt5M. We have adjusted the paragraph in the discussion to make this clearer.

Apart from SUMO2 also other Ubls, such as Nedd8 and ISG15 have been shown to form Ub/Ubl hybrid chains¹¹. In lack of knowledge on enzymes that generate these hybrid chains and in absence of a robust and easy methodology to chemically access some Ubl proteins, their biological roles remain cryptic. Importantly, we have shown before that sortylation is compatible with other Ubls as e.g. SUMO18. We therefore envision that Ubl-tools can be easily applied to all different Ubls by introducing the Srt2A/Srt5M recognition motifs into their

respective C-termini (leading to just one or two point mutations), constituting thereby a valuable tool for generating these otherwise inaccessible Ub/Ubl topologies.

REVIEWERS' COMMENTS

Reviewer #1 (Remarks to the Author):

This reviewer is enthusiastic about the publication of this manuscript. I commend the authors on their responses to the critique.

Reviewer #2 (Remarks to the Author):

In the revised manuscript, the authors have adequately addressed the previous concerns by providing new data including recognition of di-Ubs derived from the Ub mutants by UBDs. The novel technology reported here and detailed information regarding the properties of these Ub/Ubl materials are valuable for future researches in this field. The paper is now suitable for acceptance.

Reviewer #3 (Remarks to the Author):

Fottner et al. have addressed all the comments of the reviewers in a detailed point-by-point reply. The additional data and extra discussion provided by the authors have complemented the manuscript. Overall, the described Ubl-tool technique can provide valuable opportunities for studying the functional impact of these complex type of modifications. I hereby support its publication in Nature Communications.